# DEFT: Decompositional Efficient Fine-Tuning for Text-to-Image Models

Komal Kumar[1*]   Rao Muhammad Anwer[1]   Fahad Shahbaz Khan[1]
Salman Khan[1]   Ivan Laptev[1]   Hisham Cholakkal[1]

[1]Mohamed bin Zayed University of Artificial Intelligence Abu Dhabi, UAE

{komal.kumar, rao.anwer, fahad.khan,
salman.khan, ivan.laptev, hisham.cholakkal}@mbzuai.ac.ae

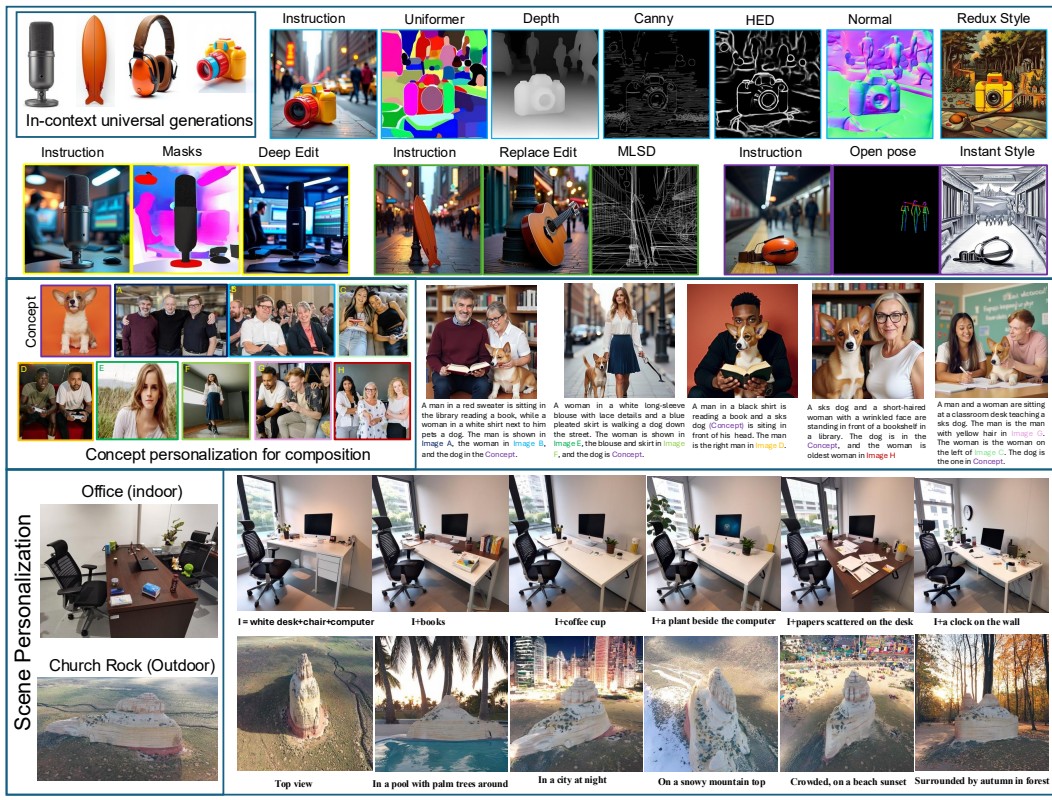

Figure 1: This figure illustrates the tasks achieved using our Decompositional Efficient Fine-Tuning (DEFT) method across diverse settings. It includes in-context learning for a variety of image generation tasks, concept personalization for composition, where these compositions emphasize the personalization of the environment and the interaction between characters and pets, blending various scenarios. Moreover, it highlights outdoor and indoor scene personalization, featuring Church Rock and the office. All the tasks fine-tune the baseline OmniGen model [48] using DEFT.

## Abstract

Efficient fine-tuning of pre-trained Text-to-Image (T2I) models involves adjusting the model to suit a particular task or dataset while minimizing computational

---

[*]Corresponding author.

39th Conference on Neural Information Processing Systems (NeurIPS 2025).

resources and limiting the number of trainable parameters. However, it often faces challenges in striking a trade-off between aligning with the target distribution: learning a novel concept from a limited image for personalization and retaining the instruction ability needed for unifying multiple tasks, all while maintaining editability (aligning with a variety of prompts or in-context generation). In this work, we introduce DEFT, Decompositional Efficient Fine-Tuning, an efficient fine-tuning framework that adapts a pre-trained weight matrix by decomposing its update into two components with two trainable matrices: (1) a projection onto the complement of a low-rank subspace spanned by a low-rank matrix, and (2) a low-rank update. The single trainable low-rank matrix defines the subspace, while the other trainable low-rank matrix enables parameter adaptation within that subspace. We conducted extensive experiments on the Dreambooth and Dreambench Plus datasets for personalization, the InsDet dataset for object and scene adaptation, and the VisualCloze dataset for a universal image generation framework through visual in-context learning with both Stable Diffusion and a unified model. Our results demonstrated state-of-the-art performance, highlighting the emergent properties of efficient fine-tuning. Our code is available on DEFT.

# 1 Introduction

Text-to-image (T2I) models have revolutionized the way we generate images, transforming abstract concepts and textual descriptions or prompts into compelling visual representations. Models such as Stable Diffusion have gained significant attention for their ability to generate high-quality images across various domains. Despite these impressive capabilities, adapting T2I models to specific tasks—such as personalization for novel concepts [34, 31] or multi-task generalization [25]—remains a challenge. Traditional fine-tuning approaches often require substantial computational resources and large amounts of reference data to train or adapt the model [19], which limits their practicality for many applications. This limitation is particularly evident in tasks requiring personalization, such as adapting a model to generate images of a specific subject or scene, where training or reference data is often scarce. For instance, Custom Diffusion [19] tends to overfit when trained on a limited number of reference images, leading to poor generalization across prompts [45, 34].

DreamBooth [35] introduced a personalized image generation method in which Low-Rank Adaptation (LoRA) [17] was employed to enable subject-specific tuning by applying learned LoRA modules at inference time, thereby preserving the identity of the subject during image generation. Although DreamBooth and other LoRA-based methods, such as LoraMerge [6] and ComposLoRA [56], improve subject personalization and separation using a limited number of reference images, they still lack fine-grained control over pose, spatial positioning, and lighting. These methods also often struggle with convergence and overfitting, especially when the low-rank updates are unconstrained and do not align well with the pre-trained weights. Furthermore, many existing LoRA-based approaches suffer from limited instruction following capabilities, particularly in complex scenarios involving multiple subjects or scenes (See the Figure 1).

In this work, we introduce DEFT (Decompositional Efficient Fine-Tuning), a novel fine-tuning framework that overcomes these limitations by decomposing the weight matrix update into two components: (1) a projection onto the orthogonal complement of a low-rank subspace, and (2) a low-rank update that allows for flexible adaptation within that subspace. This decomposition enables more efficient adaptation of pre-trained models to new tasks without sacrificing the model's ability to generalize to a wide range of scenarios. By using two trainable low-rank matrices, DEFT extends the adaptability of the model while maintaining stability during the adaptation process, ensuring that the model doesn't suffer from catastrophic forgetting.

We evaluate DEFT on several challenging datasets, including Dreamboot [34] and Dreambench Plus [31] for personalization, the InsDet [37] dataset for object and scene adaptation, and the Visual-Cloze [25] dataset for universal image generation via visual in-context learning. Our experiments demonstrate that DEFT achieves better performance than state-of-the-art methods, highlighting its effectiveness and flexibility in fine-tuning models for diverse tasks. In particular, DEFT excels in tasks such as multi-concept composition, expanding model capabilities to universal image generation [25], and reducing overfitting on small datasets. Moreover, we show that DEFT offers significant improvements in personalization, especially when adapting models to multiple subjects or complex

scenes [37], overcoming issues such as blending artifacts and limited control over pose and context. Unlike other methods, DEFT overcomes challenges such as blending artifacts and offers finer control over pose and context. Using its low-rank update in subspace, DEFT maintains a balance between efficient adaptation and generalization, providing greater flexibility in fine-tuning while minimizing the risk of catastrophic forgetting, advantages that are not fully realized by existing methods like DreamBooth or LoRA. The key contribution of the paper can be summarized as follows.

- We propose a novel fine-tuning framework, DEFT, which improve adaptability by decomposing weight updates into two components: a projection onto a low-rank subspace and a low-rank adjustment. This structure enables efficient weight editing while preserving prior knowledge.
- DEFT supports efficient fine-tuning with minimal data, enabling personalization and adaptation to new tasks without extensive retraining or risk of overfitting.
- We validate DEFT with extensive experiments on datasets such as DreamBooth, Dreambench Plus, VisualCloze, and InsDet, demonstrating its effectiveness in personalization and universal image generation. DEFT shows performance across subjects, scenes, and multi-concept compositions.

## 2 Related work

**Personalized Diffusion Models by tuning:** A simple yet effective way to adapt concepts in text-to-image models is by selectively tuning a subset of parameters, tailoring them to the desired concepts [8], [35], [7], [43], [54], [33], [55]. Methods like DreamBooth [35] fine-tune all model weights on a few images but are resource-intensive and prone to overfitting. Textual Inversion [8] is a lighter approach, learning a new token embedding to represent a concept without modifying the model. While efficient, it struggles with fine details. Custom Diffusion [19] fine-tunes a subset of parameters, learning concepts in minutes, and supports multi-concept training. Perfusion [41] and AnyHyper [2, 10] reduce trainable parameters but still face issues like overfitting and interference. These methods improve the text embedding space [8], use fine-tuning [35, 19, 17, 11, 14, 6, 28], or provide adapters [29, 53] for personalized control, with some training-free approaches [5, 10, 18, 38, 46]. LoRA [17] and SVDiff [14] decompose matrices in diffusion models, but these focus on adjusting scale rather than structure. PaRa [4] eliminates components during personalization for more stable, robust model mixing, avoiding the overfitting risks of scale adjustments [26, 13, 15]. Our dynamic framework adapts fine-tuning methods like LoRA [6] and PaRa [4], which project onto a low-rank subspace and have a flexible low-rank update for new concepts or tasks.

**Unified Models fine-tuning:** Unified [48, 25] are the single transformer-based models support image generation tasks where text and images are encoded and concatenated, separated tokens and fed to the single transformer models. These models share the same transformer blocks and can utilize objectives such as MaskGIT [3] or [27] for image generation. Unified models also support in-context learning along with a prompt. Combining multiple personalized concepts in one image is more complex than generating a single concept. Fine-tuning a model on multiple concepts can lead to interference or dominance of one concept over others [19, 12]. Early methods inserted multiple learned tokens into prompts, but diffusion models often fail to distinctly generate all concepts, especially if they share attributes or were not jointly trained. Kumari et al. [19] addressed this by fine-tuning individual concept models at low rates or merging them, allowing composition of new concepts with quality trade-offs. Mix-of-Show [12] uses a fusion LoRA and additional guidance such as sketches, improving composition, but requiring extra input and retraining. Without guidance, it suffers from concept vanishing or mixing [12]. LoRA-Composer [56] merges multiple LoRAs inference without retraining, maintaining concept isolation and visibility, demonstrating improved results in 2-3 concept compositions. Wu et al. [47] introduced the LoRA Expert Mixture (MOLE), which combines LoRA through learned gating weights, balancing multiple concepts. These methods show promise, but still require manual tuning or supervised training, and may struggle with many concepts or without guided layouts. In this work, we employ low-rank DEFT adapters to fine-tune models for new tasks, including tasks that were not originally supported by the base models.

## 3 Methodology

Figure 2 provides an overview of DEFT and compares it to other efficient fine-tuning techniques, namely PaRa [4] and LoRA [6]. We begin by considering image generation models such as Stable

Diffusion (SD) [32] and the unified Omnigen [48] model as our base architectures. SD, a latent text-to-image generation model, is based on Diffusion Probabilistic Models (DDPM) [16, 39]. Similar to LoRA's approach in diffusion models [6], we introduce low-rank adapters to the UNet and condition encoders that operate in the latent space [30]. In the case of unified models, we apply low-rank adapters to the linear layers of the transformer.

In this methodology, we first describe the foundational low-rank adaptation mechanism, then build upon this to introduce DEFT, which enhances flexibility while maintaining model generalization. The following sections explain these methods in detail.

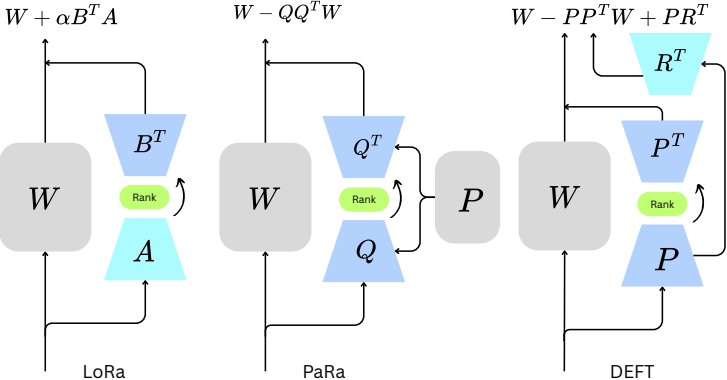

Figure 2: The figure illustrates the difference between our DEFT and other similar efficient fine-tuning techniques: (a) LoRA (Low-Rank Adaptation) modifies the pre-trained weight matrix $W$ by adding a low-rank update $\alpha B^T A$, where $A$ and $B$ are trainable low-rank matrices. (b) PaRa (Parameter-efficient Rank Adaptation) introduces a similar low-rank update, $W - QQ^T W$, with $Q$ representing the low-rank orthonormal basis. (c) DEFT (Decompositional Efficient Fine-Tuning) further decomposes the update into two components: a projection onto the subspace, $W - PP^T W$, and a low-rank adjustment, $PR^T$, where $P$ and $R$ are trainable matrices that enable more flexible adaptation while preserving the model's generalization ability.

## 3.1 Decompositional Efficient Fine-Tuning

Let $M = \{W_1, W_2, \ldots, W_n; W_i \in \mathbb{R}^{m_i \times n_i}$ pre-trained weights. Instead of updating the full weight matrix $W_i$, which can be computationally expensive requiring $O(mn)$ per matrix, the update is decomposed into smaller matrices to reduce the number of trainable parameters. We can assume a $W \in M \in \mathbb{R}^{m \times n}$ pre-trained weight matrix. We aim to compute adapted weights $W'$, such that: $W' = W + \Delta W$ where $\Delta W$ is a low-rank update constrained to rank $r \ll \min(m, n)$ so that $O(mn)$ reduce significantly. DEFT builds on the concept of low-rank adaptation, as introduced in techniques like LoRA [6] and PaRa[4]. While LoRA injects trainable low-rank matrices into specific layers, our approach goes further by decomposing the low-rank update into two components: the projection onto a subspace and the low-rank adjustment. This decomposition allows us to adapt the model more flexibly to a broader range of tasks while preserving its original performance. LoRA injects trainable low-rank matrices into specific layers while freezing the original weights. For a linear layer with input $x \in \mathbb{R}^n$, the modified forward pass becomes: $h = Wx + \Delta Wx = Wx + BAx$, $A$ is initialized with random Gaussian weights, and $B$ with zeros, ensuring $\Delta W = 0$ at initialization.

The efficacy of low-rank updates is grounded in the observation that neural networks reside on low intrinsic manifolds during adaptation [1]. For a pre-trained weight $W$, the update $\Delta W$ amplifies task-specific feature directions not emphasized in $W$. Let $U\Sigma V^\top$ be the SVD of $W$. The projection of $\Delta W$ onto $W$'s singular vectors reveals that $\Delta W$ primarily modifies directions orthogonal to $W$'s dominant singular vectors.

Inspired by this orthogonal direction, PaRa [4] introduced the low rank update in the orthogonal complement of the freeze weight, and they used rank reduction via the following formulation: $W = W_0 - QQ^T W_0$, where $Q$ is an orthonormal basis matrix and $W_0$ is the pre-trained weight matrix (Please refer to Appendix B for the proof). This operation reduces the rank of the weight matrix

$W_0$, ensuring that the model's adaptation is constrained to a smaller and more relevant subspace for the target task.

We extend the idea of modifying subspace with the low-rank update to improve the flexibility for diverse tasks in DEFT. The column space of $W_{\text{total}} = (I - QQ^\top)W_0 + QR$ extends the subset of $W_0$'s column space by incorporating $QR$(For the full proof, please refer to Appendix C), enabling adaptation to new tasks which can be described as a decomposition in trainable matrices Appendix D as:

$$W_{\text{total}} = (I - QQ^\top)W_0 + QR \tag{1}$$

By adding $QR$ to $W_{\text{reduce}}$, the total column space $\text{col}(W_{\text{total}})$ becomes:

$$\text{col}(W_{\text{total}}) = \text{col}(W_{\text{reduce}}) + \text{col}(QR) \subseteq \text{col}(W_0) + \text{col}(Q).$$

If $\text{col}(Q) \nsubseteq \text{col}(W_0)$, the subspace is *extended*, allowing adaptation to new tasks. That can be seen as a low-rank update, where low-rank updates inject task-specific directions into the weight matrix. .

### 3.1.1 Decomposition approaches

To generalize the rank-reduction mechanism beyond QR decomposition, we extend our method to support several decomposition techniques for constructing the projection matrix $P \in \mathbb{R}^{d \times d}$ used to eliminate subspaces from the original weight matrix $W_0$. In QR decomposition, $B$ is decomposed as $B = QR$, where $Q$ is orthonormal, and $R$ is upper triangular. TruncatedSVD [21] uses $B = USV^\top$, where $U$ contains the top $r$ left singular vectors, and $S$ is a diagonal matrix of singular values. Low-rank matrix factorization (LRMF) [51] uses a scaled basis $\tilde{U} = U\sqrt{S}$, while non-negative matrix factorization (NMF) [23] decomposes $B$ into non-negative matrices $B \approx WH$. Eigen decomposition computes $BB^\top \in \mathbb{R}^{d \times d}$ and performs eigen decomposition to form the projection matrix $P = V_r \Lambda_r V_r^\top$. In all cases, the projection matrix follows the form $W_{\text{reduce}} = W_0 - PP^T W_0 + PR$, with the linear transformation. The final DEFT update equation becomes:

$$h = W_{\text{reduce}}x = W_0 x - PP^T W_0 x + PRx. \tag{2}$$

As illustrated in Figure 3, when using $PP^\top$ the update removes components aligned with the entire span of $P$, producing strong displacements along both positive and negative directions. By contrast, replacing it with $\text{ReLU}(P)\text{ReLU}(P)^\top$ eliminates only the non-negative portion of the span, leading to sparser and more structured updates. This connects naturally to NMF, where non-negativity encourages additive feature combinations.

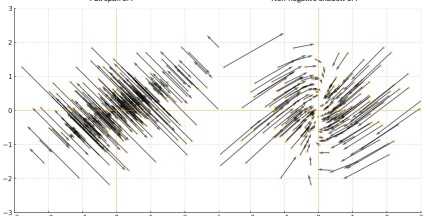

Figure 3: Displacement field visualization of the DEFT update. Left: subtraction using the full span of $P$, where both positive and negative entries contribute to the removed subspace. Right: subtraction using only the non-negative shadow $\text{ReLU}(P)$, which restricts removal to additive components and yields weaker, more selective displacements.

To further stabilize optimization, the framework applies a higher learning rate to $R$ compared to $P$, mirroring the design of LoRA. This modular formulation enables different structural biases to be introduced in a plug-and-play manner during fine-tuning, potentially improving adaptability under different data regimes or downstream tasks. To update on the framework works with the increased learning rate for $R$ compared to $P$, so this turns out to follow a similar path as LoRA. This modular formulation allows different structural biases to be introduced in a plug-and-play fashion during fine-tuning, potentially improving adaptability under different data regimes.

## 4 Experiments

### 4.1 Experimental details

To check the effectiveness of DEFT, we conducted extensive experiments. We wrote the library of the DEFT using Torch, which can be adapted to any model. We follow the similar design choice like LoRA [17]. DEFT aim to inject knowledge through low-rank updates while maintaining the

original model's stability. We use rank ($r$) equal to 4 for DreamBench Plus [31] benchmarking for all of the methods. For Visual cloze [25] universal image generation, we used $r = 32$. For evaluation purpose, we conducted experiments on universal image generation from Visualcloze [25], image personalization on DreamBoot [34], subject (live, no-living, and styles) personalization, and scene personalization. Please see the Figure 1, for the tasks supported with qualitative results. All experiments were conducted with 4 NVIDIA RTX A6000 50GB GPUs.

**Baseline:** We extend stable diffusion and stable diffusion XL (SDXL) [32] image personalization, along with Omnigen [48], into a unified model. This model utilizes a single transformer-based image generation framework, simultaneously modeling text tokens and image tokens for multi-concept composition. The goal is to expand the model's capabilities to support the DEFT module, enabling subject-wise parameter tuning and controlled scene generation without requiring full model fine-tuning. For training and evaluation, we use the diffuser library [42].

**Dataset:** For evaluation, we use several datasets, including the DreamBooth Benchmark [34], which contains 30 personalized concepts across 15 categories, with 4–6 images per concept and 25 challenging prompts. Additionally, we curate datasets for single-subject and multi-subject personalization, ensuring diverse compositions. Key datasets include VisualCloze [25] (We created 3M instructions for in-context learning for fine-tuning), DreamBooth (30 concepts for subject-driven generation), DreamBench Plus [31] (150 concepts for human-aligned benchmarks), and InsDet [37], the high-resolution dataset for instance detection with 100 objects and 5 scenes. We created instructions for these 100 objects with an average of 22+ images using Blip2 [24], and for the scene, we used Qwen 32B [50] for the details description.

## 4.2 Results and Analysis

### 4.2.1 DEFT improves Instruction following abilities

Table 1 reports CLIP-T scores, measuring image–text alignment across T2I models. Traditional methods (Textual Inversion, DreamBooth) and recent variants (LoRA, BLIP-Diffusion, PaRa) show varying performance on SD v1.5 and SDXL. Our DEFT, built on SDXL v1.0, achieves the best score (0.361), surpassing LoRA (0.341) and PaRa (0.354). This improvement stems from DEFT's low-rank injection, which expands the fine-tuning subspace, retaining the original model's instruction-following capabilities, enabling more coherent image generation in personalization tasks.

Table 1: This table illustrates the performance of several T2I models on 150 subjects with eight different and diverse prompts from the Dreambench Plus [31] dataset. We evaluated with their associated CLIP-T scores used for image-text alignment. The results show the ability of each model to integrate visual and textual information across different versions of models, such as SD v1.5 and SDXL v1.0. For our DEFT, we used the same setting as DreamBooth LoRA [6], including the SDXL.

| Frameworks | T2I Model | CLIP-T |
|---|---|---|
| Textual Inversion [9] | SD v1.5 | 0.302 |
| DreamBooth [34] | SD v1.5 | 0.323 |
| DreamBooth LoRA [6] | SDXL v1.0 | 0.341 |
| BLIP-Diffusion [24] | SD v1.5 | 0.286 |
| Emu2 [40] | SDXL v1.0 | 0.310 |
| IP-Adapter-Plus [52] ViT-H | SDXL v1.0 | 0.282 |
| IP-Adapter [52] ViT-G | SDXL v1.1 | 0.309 |
| PaRa [4] | SDXL v1.0 | 0.354 |
| DreamBooth DEFT (Ours) | SDXL v1.0 | **0.361** |

Table 2: Style transfer and conditional generation comparison for quantitative performance on Visualcloze [25] test dataset. In Visualcloze results, FLUX.1dev [20] is represented as dev and FLUX.1-Fill-dev as Fill. Both DEFT are evaluated on rank = 32 fine-tuned with the OmniGen [48] model. Different scores are used for the two distinct tasks.

| Condition | Method | CLIP-Score | Image Consistency | |
|---|---|---|---|---|
| | | Image | DINOv1 | DINOv2 |
| Canny | OmniGen | 95.45 | 87.13 | 87.60 |
| | VisualCloze | 89.32 | – | – |
| | **DEFT (Ours)** | **95.78** | **90.37** | **90.65** |
| Depth | OmniGen | 92.02 | 85.16 | 77.39 |
| | VisualCloze | 87.56 | – | – |
| | **DEFT (Ours)** | **93.18** | 88.98 | **85.75** |
| **Style Type** | **Method** | **Text Score($\uparrow$)** | **Image Score($\uparrow$)** | **F1($\uparrow$)** |
| InstantStyle | InstantStyle [44] | 0.27 | 0.60 | |
| | OmniGen [48] | 0.27 | 0.52 | 0.55 |
| | VisualCloze-dev [25] | 0.30 | 0.53 | |
| | VisualCloze-fill [25] | **0.29** | 0.55 | |
| | **DEFT (Ours)** | 0.28 | **0.69** | **0.59** |
| ReduxStyle | OmniGen [48] | 0.27 | 0.58 | 0.47 |
| | VisualCloze-dev [25] | **0.29** | 0.53 | |
| | VisualCloze-fill [25] | 0.27 | 0.55 | |
| | **DEFT (Ours)** | 0.26 | **0.69** | **0.49** |

### 4.2.2 Universal Image Generation through Adapter

Beyond personalization, we assess DEFT's generalization by fine-tuning OmniGen [48] on Visual-Cloze [25]. Tables 2 and 3 show that DEFT consistently outperforms OmniGen and VisualCloze in both style transfer and conditional generation.

For image consistency, DEFT yields higher CLIP and DINO scores under Canny Edge and Depth Map conditions, demonstrating stronger alignment between text and visuals. In style transfer, it achieves the best Image Scores (0.69) and F1 metrics across both InstantStyle and ReduxStyle tasks, indicating better preservation of textures and styles.

Table 3 further highlights DEFT's advantage in controllability and quality. Under Canny Edge, DEFT achieves the highest F1 and SSIM, with competitive FID. Under Depth Map, it maintains strong controllability (F1 = 0.86) while substantially improving SSIM (0.72). These results confirm that DEFT enhances both structural fidelity and perceptual quality, making it well-suited for universal image generation.

Table 3: Comparison of various methods on controllability and quality metrics across Canny Edge and Depth Map conditions. Our DEFT fine-tune OmniGen [48].

| | Canny Edge | | | | Depth Map | | | |
| | Controllability | | Quality | | Controllability | | Quality | |
| Method | F1 ↑ | RMSE ↓ | FID ↓ | SSIM ↑ | F1 ↑ | RMSE ↓ | FID ↓ | SSIM ↑ |
| --- | --- | --- | --- | --- | --- | --- | --- | --- |
| ControlNet [53] | 0.13 | - | 46.06 | 0.34 | 0.13 | 23.7 | 36.83 | 0.41 |
| OmniControl [49] | 0.47 | - | 29.58 | 0.61 | 0.47 | 21.44 | 36.23 | 0.52 |
| OneDiffusion [22] | 0.39 | - | 32.76 | 0.55 | 0.39 | 39.03 | 39.03 | 0.49 |
| OmniGen | 0.43 | 4.55 | 51.58 | 0.47 | 0.85 | 9.83 | 115.54 | 0.69 |
| VisualCloze_dev | 0.39 | - | 30.36 | 0.61 | 0.39 | 25.06 | 42.14 | 0.53 |
| VisualCloze_Fill | 0.35 | - | **30.6** | 0.55 | 0.35 | 10.31 | **33.88** | 0.54 |
| **DEFT (Ours)** | **0.48** | **4.11** | 46.62 | **0.66** | **0.86** | **9.38** | 46.74 | 0.72 |

### 4.2.3 Qualitative comparison

In Figure 4, we compare the consistency of reference images across various tasks from DreamBench Plus [31]. The DEFT method outperforms LoRA in generating consistent, high-quality images. DEFT effectively maintains visual coherence and fine details, such as textures and proportions, when adapting to diverse prompts involving cat, horse, and pixel warrior. For example, the horse color is maintained in most of the images compared to LoRA.

In Figure 5, we showcase various objects and live subjects from Dreambench Plus [31]. LoRA and our DEFT shows comparable results but there are some details that can seen. For instance, in the Jellyfish Image, DEFT captures delicate transparency and lighting, which PaRa and LoRA fail to replicate. Similarly, in the Taxi Image, DEFT preserves sharp details like streetlights and reflections, while the other methods show blurred elements. The Drum Image demonstrates DEFT's ability to maintain fine textures and lighting, unlike PaRa and LoRA, which produce less-defined results. In the Pokémon Figures Image, DEFT preserves vibrant colors and detailed textures, while PaRa and LoRA produce duller, less detailed images. Lastly, DEFT excels at reproducing fine fur textures and eye clarity in the Cat Image, where PaRa and LoRA struggle with detail preservation.

### 4.2.4 Emergent properties efficent-finetuning

The emergent properties of efficient fine-tuning are evident when a model, initially unaware of specific concepts, is trained on a small set of images representing diverse scenarios. In the Figure 6, we show the multi-concept personalization by finetuning omnigem, which does not require a separate LoRA adapter for each task like LoRAMerge [6]. In the first row, the model is fine-tuned on a few pet-related images, such as a Corgi dog and a teddy bear, in various environments. It learns to recognize key features of pets, like shape, color, and behavior, despite having no prior knowledge. In the second row, the model demonstrates its ability to generalize these learned features, successfully recognizing pets in new contexts—such as outdoors or with people—under different lighting and settings. The third row shows the model fine-tuned on a small number of object-related images, including toys and accessories. Here, the model learns to identify objects based on features like shape,

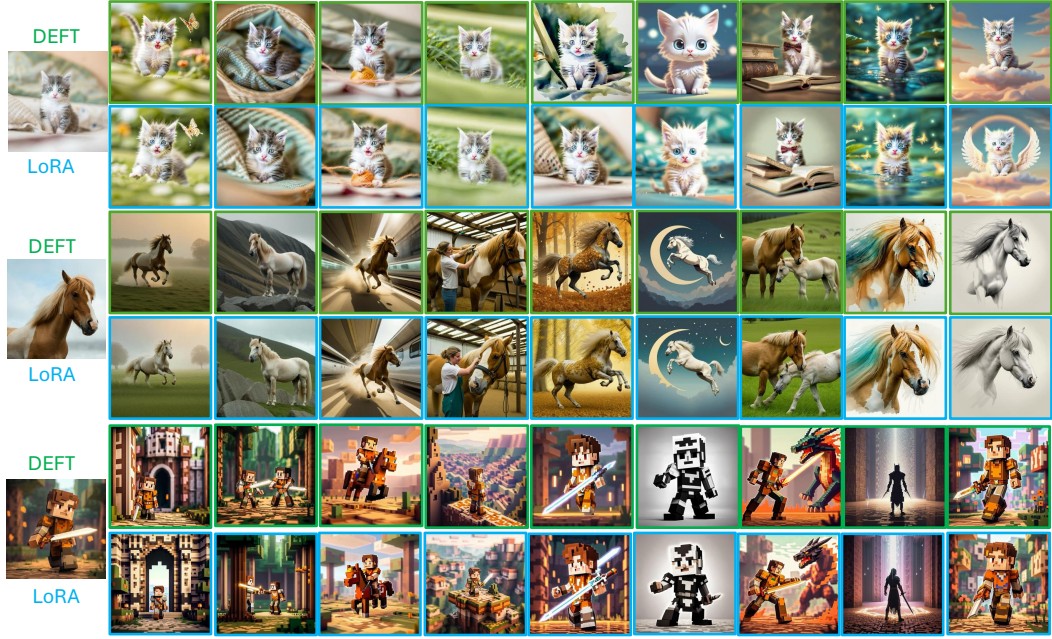

Figure 4: Comparison of image generation consistency between our DEFT and LoRA across various categories to check the consistency of the reference. The images show results for different subjects, including cats, horses, and pixel worrier (Zoom for best view).

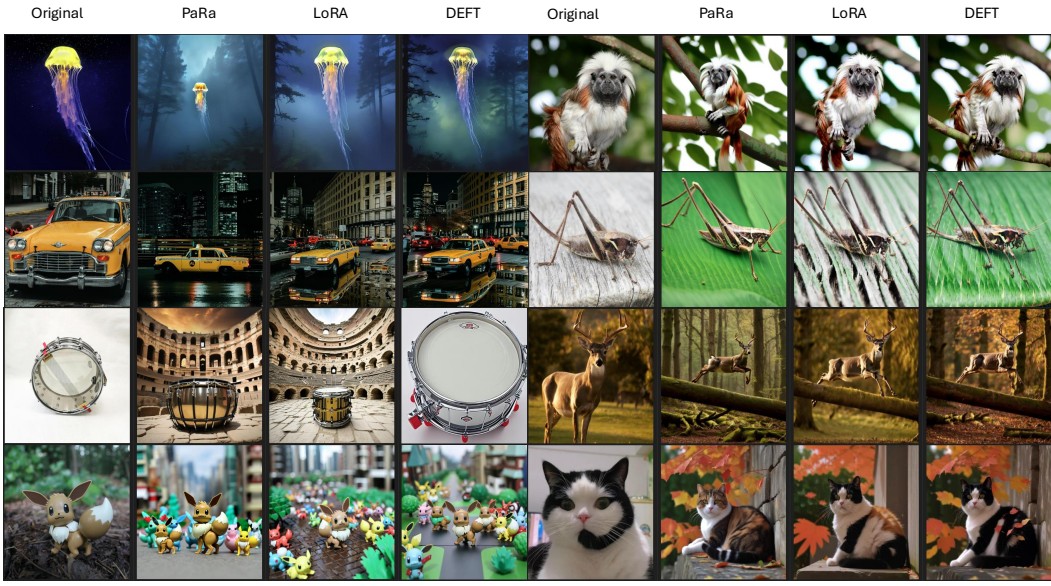

Figure 5: This figure compares the performance of LoRA, PaRa, and DEFT models in object-based tasks using the DreambenchPlus dataset. All methods use the same prompt, with the ideal model exhibiting results that closely resemble the original while maintaining diversity.

size, and texture. In the fourth row, the model's generalization ability is further tested as it accurately recognizes these objects in new, unseen conditions, such as backpacks in varied environments like parks, streets, and cityscapes. This illustrates how efficient fine-tuning, with minimal data, empowers the model to not only learn specific concepts but also extend its knowledge to handle a wide array of tasks and environments without requiring extensive retraining.

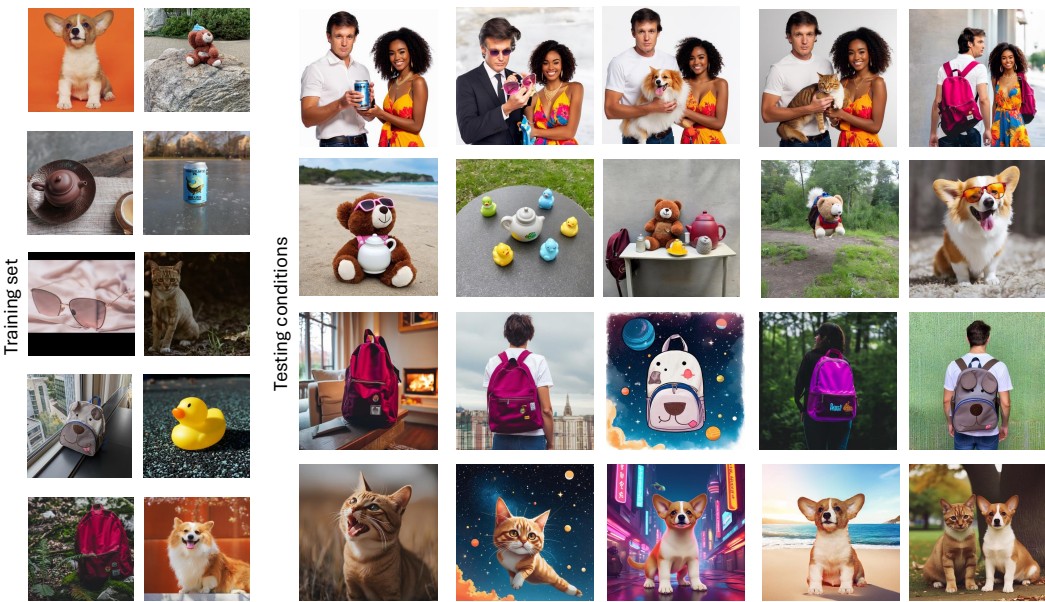

Figure 6: This figure demonstrates the emergent properties of the model, which adapts the training set to generate diverse combinations under varying testing conditions. The model successfully generates new and creative image variations by combining elements from different objects and scenes, showcasing its ability to generalize beyond the original training data and produce novel compositions. All the results on the right side were obtained after fine-tuning OmniGen [48] using DEFT.

## 4.3 Ablation study

To better understand DEFT's behavior, we include ablations on decomposition methods, training duration, and novel capabilities.

**Effect of decomposition.** In Sec. 3.1.1, we described the various decomposition methods that can function the eqn 1 other than QR decomposition. In Figure 7, we provide a visual example of different decompositions on fine-tuning in the DEFT style. All methods, including LRMF, NMF, QR, TSVD, and Relaxing P, closely mimic the reference image, each capturing key aspects such as lighting, subject clarity, and environmental context. In the default setting of our DEFT, we used it without decomposition because this configuration provides a baseline for evaluating the performance of the model with minimal modifications. By avoiding decomposition initially, we ensure that any observed improvements or changes in performance can be directly attributed to the fine-tuning process itself, rather than the additional complexity introduced by matrix decompositions.

**Decomposition Methods**. As shown in Table 4 and Figure 7, alternative decompositions (e.g., NMF, QR, TSVD) achieve competitive performance, with learnable variants (Relaxing P, $P_{nmf}$) providing stronger prompt control. This highlights that DEFT is flexible to different factorization strategies, though our default setting avoids decomposition for simplicity.

Table 4: Performance comparison of different decomposition methods as discussed in section 3.1.1. Relaxing P and $P_{nfm}$ indicates the learnable matrix which directly adapts the $W_0$ as shown in the Eq 2. $P_{nfm}$ is a non-negative version for adaptation as shown in the Figure 3. All decomposition approaches demonstrate competitive performance, with $P_{nfm}$ showing significant control over prompts, highlighting the value of structural non-negativity as a fine-tuning bias. All results are evaluated with DreamBooth [34] Dog (See Figure 7) with 8 diverse prompts.

| Method | Speed (ms) | CLIP-I | CLIP-T | DINO-V1 | Aesthetic | Sharpness |
|---|---|---|---|---|---|---|
| LRMF | 29.10 | $0.827 \pm 0.038$ | $0.220 \pm 0.051$ | $0.307 \pm 0.040$ | $0.014 \pm 0.005$ | $349 \pm 414$ |
| NMF | 5.16 | $0.883 \pm 0.033$ | $0.222 \pm 0.042$ | $0.357 \pm 0.068$ | $0.015 \pm 0.003$ | $206 \pm 91$ |
| QR | 5.38 | $0.875 \pm 0.043$ | $0.217 \pm 0.041$ | $0.340 \pm 0.060$ | $0.017 \pm 0.003$ | $215 \pm 106$ |
| TSVD | 28.72 | $0.875 \pm 0.031$ | $0.223 \pm 0.041$ | $0.333 \pm 0.062$ | $0.016 \pm 0.004$ | $331 \pm 134$ |
| Relxing P | 5.22 | $0.879 \pm 0.030$ | $0.235 \pm 0.041$ | $0.330 \pm 0.058$ | $0.015 \pm 0.003$ | $340 \pm 150$ |
| Relexing $P_{nmf}$ | 5.22 | $0.923 \pm 0.037$ | $0.266 \pm 0.033$ | $0.440 \pm 0.096$ | $0.016 \pm 0.003$ | $175 \pm 25$ |

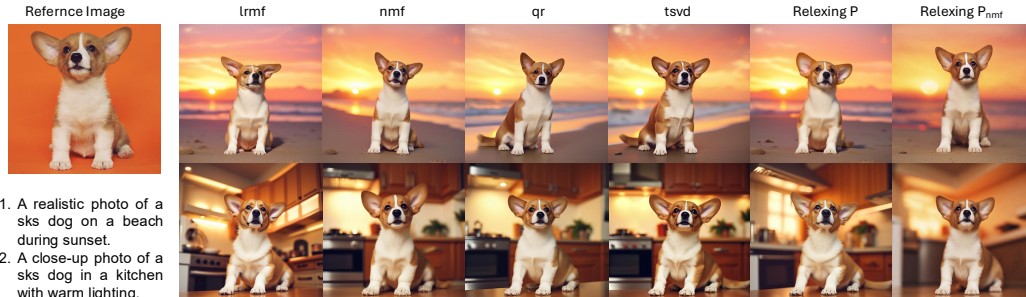

| | Refernce Image | lrmf | nmf | qr | tsvd | Relexing P | Relexing $P_{nmf}$ |

1. A realistic photo of a sks dog on a beach during sunset.
2. A close-up photo of a sks dog in a kitchen with warm lighting.

Figure 7: This figure showcases various rank-reduction techniques for personalization. By extending traditional decomposition methods such as QR, Truncated SVD, LRMF, and NMF, Relaxing P enables the projection matrix P to adapt and refine the original weight matrix $W_0$ (See Eq 2), effectively generating new image concepts without any decomposition required.

**Training Steps.** Table 5 compares DEFT with LoRA at different training horizons. While both improve with more steps, DEFT maintains stable instruction adherence (CLIP-T) and image quality, whereas LoRA suffers degraded alignment at longer training. This makes DEFT more suitable for extended fine-tuning.

Table 5: Effect of training steps on fine-tuning OmniGen with DreamBooth's Dog [34]. We compare DEFT (using QR decomposition with rank 8) against LoRA (also with rank 8). DEFT maintains a balance between image quality and instruction-following capability across training durations, whereas LoRA suffers a significant drop in instruction adherence, especially at longer training horizons.

| | Training Steps = 2000 | | | | Training Steps = 8000 | | | |
| Method | CLIP-I | CLIP-T | Aesthetic | Sharpness | CLIP-I | CLIP-T | Aesthetic | Sharpness |
|---|---|---|---|---|---|---|---|---|
| DEFT | $0.811 \pm 0.066$ | $0.302 \pm 0.026$ | $0.015 \pm 0.005$ | $294 \pm 261$ | $0.882 \pm 0.059$ | $0.319 \pm 0.025$ | $0.015 \pm 0.003$ | $320 \pm 364$ |
| LORA | $0.836 \pm 0.039$ | $0.286 \pm 0.033$ | $0.016 \pm 0.004$ | $449 \pm 334$ | $0.876 \pm 0.024$ | $0.219 \pm 0.023$ | $0.013 \pm 0.002$ | $65 \pm 3$ |

**Camera-Aware Generation.** To test whether fine-tuning can add new abilities beyond style or subject adaptation, we evaluate DEFT on 3D-aware generation using SFM data [36] extracted from a drone video of a church (see the last row of Figure 6). We trained OmniGen with all 11 intrinsic and extrinsic camera parameters using a special token. Table 6 compares pretrained, instruction-only ($DEFT_{INS}$), and camera-augmented ($DEFT_{SFM}$) inference. $DEFT_{SFM}$ shows slight but consistent gains in CLIP-I, CLIP-T, and Sharpness over both baselines, suggesting that DEFT can be extended toward camera-aware generation. A camera position encoding could further improve this capability.

**Efficiency.** Furthermore, we analyzed training efficiency for the rank-64 configuration with a batch size of 2 on the OmniGen 3.762B parameter model. LoRA achieved 12 steps/sec, as it does not require decomposition or matrix multiplication, with a peak memory usage of 8.03 GB and 37.7M trainable parameters. DEFT ran at 11 steps/sec with 7.95 GB peak memory and the same 37.7M trainable parameters; even with relaxed $P$ (no decomposition), matrix multiplication is still required. PaRa operated at 8 steps/sec with a peak memory usage of 7.86 GB and 25.2M trainable parameters

Table 6: Testing DEFT's camera-aware generation on SFM data: pretrained vs. instruction-only vs. instruction+camera parameters.

| | CP-I | CP-T | DO-V1 | Sharp |
|---|---|---|---|---|
| Pre | 0.475 | 0.201 | 0.220 | 1179 |
| $DEFT_{INS}$ | 0.647 | 0.210 | 0.350 | 1813 |
| $DEFT_{SFM}$ | **0.660** | **0.213** | 0.343 | **1852** |

## 5 Conclusion

In this work, we have introduced DEFT, a novel framework designed to optimize the fine-tuning process of pre-trained models. DEFT improves the model's adaptability by decomposing weight updates into two key components: a projection onto a low-rank subspace and a flexible low-rank update. This decomposition allows the model to fine-tune efficiently, maintaining high performance while reducing the computational burden. Using DEFT, we have shown that fine-tuning can be efficient without sacrificing flexibility or model generalization. This work highlights the potential of DEFT to overcome the challenges of personalization and adaptability in text-to-image generation models. Additionally, our results suggest that DEFT could be a future fine-tuning framework.

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

# A Preliminaries and Notation

Let $W_0 \in \mathbb{R}^{m \times n}$ denote a fixed (pretrained/frozen) weight matrix. Let $Q \in \mathbb{R}^{m \times r}$ with $r \ll m$ denote a matrix whose columns are *orthonormal*, i.e., $Q^\top Q = I_r$. We write $P_Q := QQ^\top \in \mathbb{R}^{m \times m}$ for the orthogonal projector onto the subspace $\mathrm{span}(Q) \subseteq \mathbb{R}^m$; then $I - P_Q$ is the projector onto $\mathrm{span}(Q)^\perp$.

We use $\mathrm{col}(A)$ for the column space (image) of a matrix $A$. For two subspaces $\mathcal{U}, \mathcal{V} \subseteq \mathbb{R}^m$, the *sum* $\mathcal{U} + \mathcal{V} := \{u + v : u \in \mathcal{U}, v \in \mathcal{V}\}$ and $\dim(\mathcal{U} + \mathcal{V}) = \dim \mathcal{U} + \dim \mathcal{V} - \dim(\mathcal{U} \cap \mathcal{V})$.

We define the *reduced* baseline $W_{\text{reduce}} := (I - P_Q)\, W_0 = (I - QQ^\top)\, W_0$, and the *adapted* total weight $W_{\text{total}} := W_{\text{reduce}} + QR = (I - QQ^\top)W_0 + QR$, where $R \in \mathbb{R}^{r \times n}$ is trainable (low rank through $r$).

# B Proof of Column Space Subset

**Claim:** The column space (image) of $W_{\text{reduce}} = W_0 - QQ^\top W_0$ is a subset of the column space of $W_0$.

**Proof:** Let $S_0 = \mathrm{col}(W_0)$, the column space of $W_0$ and $Q$ be an orthogonal matrix from the QR decomposition of $P$, so $Q^\top Q = I$.

Now any vector $v \in \mathrm{col}(W_{\text{reduce}})$ can be written as:

$$v = W_{\text{reduce}}x = (IQQ^\top)W_0 x,$$

where $x$ is an arbitrary input vector.

To decompose $W_0 x$, let $y = W_0 x$. By definition, $y \in S_0$. The term $QQ^\top y$ is the orthogonal projection of $y$ onto $\mathrm{col}(Q)$. If $\mathrm{col}(Q) \subseteq S_0$, then $QQ^\top y \in S_0$, since projections onto subspaces of $S_0$ remain in $S_0$.

First, we will check the Linearity of $S_0$. Since $S_0$ is a subspace, it is closed under addition and scalar multiplication. Therefore, $yQQ^\top y \in S_0$, as both $y$ and $QQ^\top y$ are in $S_0$. Every $v = (IQQ^\top)y$ lies in $S_0$. Thus, $\mathrm{col}(W_{\text{reduce}}) \subseteq \mathrm{col}(W_0)$.

We assumes $\mathrm{col}(Q) \subseteq \mathrm{col}(W_0)$. This is enforced during training by:

- Initializing $P$ to zero, ensuring $Q$ starts within $S_0$.
- Fine-tuning $P$ on task-specific data, which implicitly aligns $\mathrm{col}(Q)$ with directions in $S_0$ relevant to the task.

By subtracting components of $W_0$ along learned orthonormal bases $Q$, the method reduces the output space *within* $S_0$, preserving the original model's expressivity while enabling parameter-efficient adaptation. The rank reduction is controlled by the dimensionality of $Q$, which is typically small (e.g., $r = 4$).

The column space of $W_{\text{reduce}}$ is a subset of $W_0$'s column space because $W_{\text{reduce}} = (IQQ^\top)W_0$, and the projection $QQ^\top W_0$ removes components of $W_0$ *only within* the subspace spanned by $Q$. Since $Q$ is learned to lie in $\mathrm{col}(W_0)$ during training, the difference $W_0 QQ^\top W_0$ remains entirely in $\mathrm{col}(W_0)$. This ensures dimensionality reduction without exiting the original output space, as required.

# C Extension of Column Space with $W_{\text{total}}$

**Claim:** The column space of $W_{\text{total}} = (IQQ^\top)W_0 + QR$ extends the subset of $W_0$'s column space by incorporating $QR$, enabling adaptation to new tasks.

**Proof:**

The total weight matrix $W_{\text{total}}$ is given by:

$$W_{\text{total}} = \underbrace{(IQQ^\top)W_0}_{W_{\text{reduce}}} + \underbrace{QR}_{\text{Low-rank adaptation}}.$$

The term $W_{\text{reduce}}$ projects $W_0$ onto the orthogonal complement of $Q$'s column space, as proven earlier. The term $QR$ adds a trainable low-rank component derived from $P$.

Based on the previous proof Appendix B, we know that:

$$\text{col}(W_{\text{reduce}}) \subseteq \text{col}(W_0).$$

This implies that $W_{\text{reduce}}$ retains the original model's learned features but in a reduced subspace.

Since $Q$ is orthogonal (from $P = QR$), we have the column space of $QR$; $\text{col}(QR) = \text{col}(Q)$. The column space of $Q$ depends on the learned matrix $P$. If $P$ evolves to include directions *outside* $\text{col}(W_0)$, then $\text{col}(Q) \nsubseteq \text{col}(W_0)$.

The column space of $W_{\text{total}}$ is the sum of the column spaces of $W_{\text{reduce}}$ and $QR$:

$$\text{col}(W_{\text{total}}) = \text{col}(W_{\text{reduce}}) + \text{col}(QR).$$

Substituting the results from the previous steps:

$$\text{col}(W_{\text{total}}) \subseteq \text{col}(W_0) + \text{col}(Q).$$

If $\text{col}(Q) \nsubseteq \text{col}(W_0)$, then the sum $\text{col}(W_0) + \text{col}(Q)$ is *strictly larger* than $\text{col}(W_0)$. This occurs when $P$ (and hence $Q$) is trained on task-specific data to capture *new directions* outside the original subspace of $W_0$.

LoRA parametrizes weight updates as $\Delta W = AB$, where $A$ and $B$ are low-rank. In this case, $QR$ serves the role of $\Delta W$, with $Q$ acting as orthonormal bases and $R$ as adaptable coefficients. Both methods enable adaptation by expanding the column space with low-rank updates.

**Conditions:**

- $B$ is initialized to zero but fine-tuned on task-specific data.
- Gradient updates allow $B$ (and $Q$) to explore directions beyond $\text{col}(W_0)$, breaking the initial assumption that $\text{col}(Q) \subseteq \text{col}(W_0)$.
- $B \in \mathbb{R}^{d \times r}$, with $r \ll d$, ensuring parameter efficiency while still enabling the expansion of the column space.

By adding $QR$ to $W_{\text{reduce}}$, the total column space $\text{col}(W_{\text{total}})$ becomes:

$$\text{col}(W_{\text{total}}) = \text{col}(W_{\text{reduce}}) + \text{col}(QR) \subseteq \text{col}(W_0) + \text{col}(Q).$$

If $\text{col}(Q) \nsubseteq \text{col}(W_0)$, the subspace is *extended*, allowing adaptation to new tasks. This mirrors LoRA's mechanism, where low-rank updates inject task-specific directions into the weight matrix.

## D  Decomposing weight matrix

**1. Decomposing a single column $w_i$.**

Let $w_i \in \mathbb{R}^m$ be any column of $W$ and let $P := QQ^\top$ denote the orthogonal projector onto $\text{span}(Q) \subseteq \mathbb{R}^m$.

1. *Parallel part (projection onto* $\text{span}(Q)$*).*

$$P\,w_i \;=\; Q\big(Q^\top w_i\big) \;\in\; \text{span}(Q).$$

Here $Q^\top w_i \in \mathbb{R}^r$ is the coordinate vector of $w_i$ in the basis $Q$; multiplying by $Q$ lifts those coordinates back to the ambient space $\mathbb{R}^m$.

2. *Orthogonal part (component orthogonal to* $\text{span}(Q)$*).*

$$(I - P)w_i, \qquad \text{with} \quad Q^\top (I - P)w_i = 0.$$

3. *Reconstruction.*

$$w_i \;=\; P\,w_i \;+\; (I - P)w_i.$$

Geometrically, $P w_i$ is the foot of the perpendicular dropped from $w_i$ onto $\mathrm{span}(Q)$; the residual vector $(I - P)w_i$ completes the decomposition.

## 2. Extending the idea to the whole matrix $W$.

Write $W = [\, w_1\, w_2\, \ldots\, w_n \,]$. Applying the column-wise decomposition to every $w_i$ yields

$$W = \big[\, Pw_1\, Pw_2\, \ldots\, Pw_n \,\big] + \big[\, (I - P)w_1\, (I - P)w_2\, \ldots\, (I - P)w_n \,\big]$$

$$= P\,[\, w_1\, w_2\, \ldots\, w_n \,] + (I - P)[\, w_1\, w_2\, \ldots\, w_n \,]$$

$$= PW + (I - P)W.$$

Because $P = QQ^{\top}$, the first term can be expressed more compactly:

$$PW \;=\; Q(Q^{\top}W),$$

so the final matrix decomposition is

$$\boxed{\; W \;=\; Q(Q^{\top}W) \;+\; (I - QQ^{\top})W \;}.$$

We are replacing this $Q^{\top}W$ with a trainable low rank matrix $R$ that extends flexibility to adapt for new tasks, keeping $W$ weights frozen. Hence, the final equation will become:

$$\boxed{\; W \;=\; QR \;+\; (I - QQ^{\top})W \;}.$$

# Supplementary Material: Emergent Properties of Efficient Fine-Tuning in Text-to-Image Models

🌐 **Dream Branch Plus Comparison:** `https://anonymousdreambranchplus.netlify.app`

🔗 **Omnigen-Visualcloze:** `https://anonymouscloze.netlify.app/`

🔗 **InsT Objects Qualitative Results:** `https://anonymousinstobjets.netlify.app`

## S1    Further Quantitative and qualitative results

### S1.0.1    Image generation consistency

In this section, we present a detailed comparison of various fine-tuning methods for different subjects using the CLIP-image score on the Dreambooth dataset, as shown in Table 7. The methods include the proposed DEFT, PaRa [4], and LoRA [6], alongside previous approaches such as Texture Inversion and DreamBooth. The table compares performance across three distinct subject categories: BEAR_PLUSHIE, CAT, and DOG8. Our proposed DEFT method, with a rank of 4, achieves high CLIP-image scores of 0.8339 for BEAR_PLUSHIE, 0.9280 for CAT, and 0.8721 for DOG8. Notably, PaRa with rank 4 shows slightly improved results, especially for DOG8, with a score of 0.8838. In contrast, LoRA methods, particularly with ranks 4 and 8, show lower performance scores, especially for BEAR_PLUSHIE. Compared to previous methods such as PaRa [4] and SVDIFF [14], our proposed methods, DEFT, exhibit competitive or superior results in terms of image-text alignment across all subject categories, underlining their effectiveness for multimodal image generation tasks.

| "A photo of [V]" | BEAR_PLUSHIE | CAT | DOG8 |
|---|---|---|---|
| DEFT (rank=8) (Our) | **0.8415** | **0.9504** | 0.8882 |
| DEFT (rank=4) (Our) | 0.8339 | 0.9280 | 0.8721 |
| PaRa [4](rank=4) | 0.8271 | 0.9315 | 0.8780 |
| PaRa [6] (rank = 8) | 0.8051 | 0.9467 | **0.8955** |
| LoRA [6](rank=4) | 0.7741 | 0.8057 | 0.7773 |
| LoRA [34] (rank = 8) | 0.7943 | 0.8583 | 0.8295 |
| SVDIFF [14] | 0.7818 | 0.8854 | 0.8363 |
| DREAMBOOTH [34] | 0.7921 | 0.8893 | 0.8392 |
| TEXTURE INVERSION [9] | 0.7421 | 0.8048 | 0.7432 |

Table 7: Comparison of Various Methods for Different Subjects Using Clip-Image Score on Dreambooth Dataset: The table presents the comparison of different fine-tuning methods on the Dreambooth dataset, evaluated using the CLIP-image score. It highlights the performance of the proposed DEFT, PaRa, and LoRA methods against previous approaches, including Texture Inversion and DreamBooth, across multiple subject categories like BEAR_PLUSHIE, CAT, and DOG8.

### S1.0.2    Qualitative comparison

Furthermore, the Figure 8 presents qualitative results comparing different fine-tuning strategies and DEFT applied to the Unified Omnigen model. It showcases various image generations of a dog across different environments and prompts, including a lush green field, a beach, a snowy landscape, a cityscape, a garden, and a forest. Each model—Base, LoRA, PaRa, and DEFT—produces distinct results, emphasizing how these fine-tuning methods affect image generation based on specific prompts. The outcomes demonstrate the ability of these techniques to enhance the generalization and adaptivity of the model while maintaining high-quality, realistic results. The comparisons underline the impact of efficient fine-tuning in improving the model's ability to generate diverse, accurate images across various scenarios.

Furthermore, the Figure 9, demonstrates the model's impressive ability to generalize across a wide range of unseen prompts. The image features four different outputs generated based on distinct themes: abstract, fantasy, futuristic, and historical prompts. Despite the varied nature of the inputs, the model consistently produces high-quality results, showing its adaptability to different styles

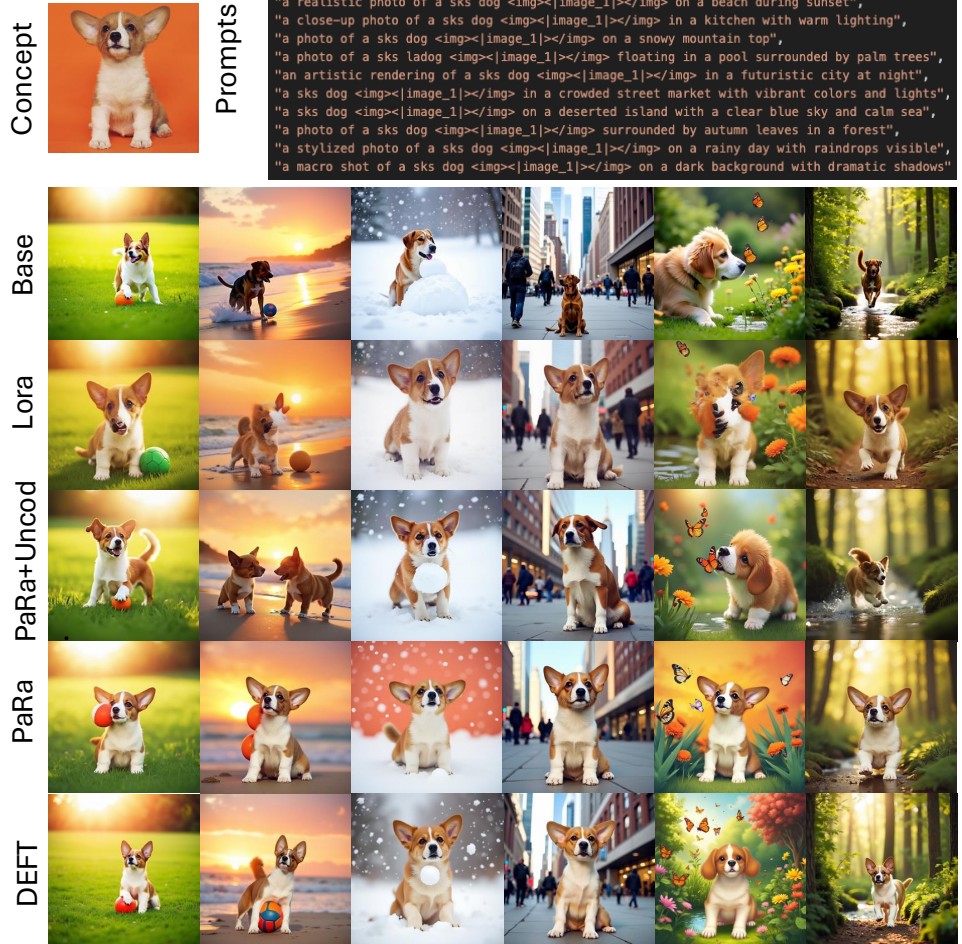

Figure 8: Qualitative Results on Unified Omnigen Model Comparing Efficient Fine-Tuning and DEFT: This figure presents qualitative results comparing efficient fine-tuning strategies and DEFT on the Unified Omnigen model. The outcomes demonstrate the capability of these techniques to enhance model generalization and adaptivity while maintaining high-quality results.

and concepts. The figure emphasizes the model's versatility, highlighting its capacity to maintain visual coherence and output quality across diverse scenarios, from abstract landscapes to historical depictions. This illustrates the robustness of the model in handling various types of prompts while ensuring consistency in the final image outputs.

### S1.0.3 Qualitative and quantitative differences

The qualitative and quantitative comparison between the methods DEFT and LoRA, as shown in both the table 8 and the images 10 on dreambench plus [31] with SDXL [32] finetuning, reveals distinct strengths for each model in generating images of cats and horses. From the quantitative perspective, LoRA consistently achieves higher scores across DINOv1 and DINOv2 for both the Kitten and HORSE images. For example, LoRA outperforms DEFT in DINOv1 and DINOv2 for the Kitten image (83.5538 vs. 79.9416 for DINOv1, and 72.2653 vs. 65.0358 for DINOv2), and similarly for the HORSE images (83.5538 vs. 77.8501 for DINOv1, and 72.2653 vs. 65.0358 for DINOv2). These results suggest that LoRA is better at capturing complex features and achieving higher-quality representations in these metrics, which might reflect its greater flexibility and artistic adaptability.

In contrast, DEFT demonstrates a stronger performance in CLIP-I and CLIP-T for some images, especially for the Kitten images (83.5867 for DEFT vs. 81.3446 for LoRA in CLIP-I), indicating its ability to produce more realistic, detailed representations that preserve the original essence of the

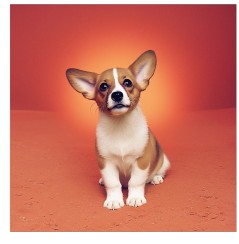 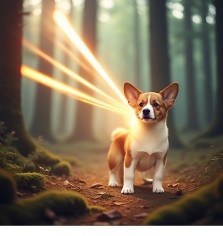 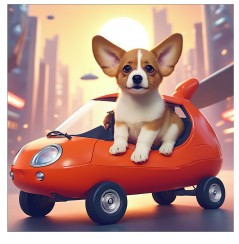 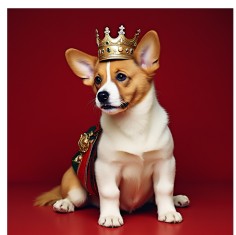

Unseen prompt abstract    Unseen prompt fantasy    Unseen prompt futuristic    Unseen prompt historical

Figure 9: Diverse Prompt Generalization: This figure shows the generalization capabilities of the model across diverse prompts, emphasizing its ability to handle a variety of inputs while maintaining consistent output quality.

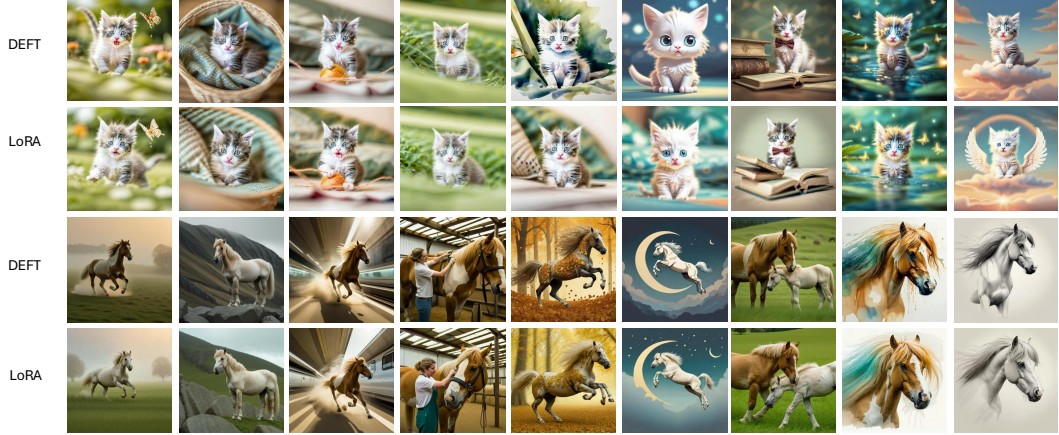

Figure 10: Qualitative comparison of the DEFT and LoRA methods for generating images of cats and horses. The first row shows images of cats, while the second row shows horses. DEFT produces realistic, detailed images, while LoRA introduces more artistic and stylized elements, showcasing its flexibility in adapting to creative representations.

animals. DEFT's output tends to have clearer textures, sharper details, and more lifelike features, showcasing its strength in realism and faithful reproduction.

The images generated by DEFT are more grounded in reality, with clear textures and natural settings. In the case of the HORSE images, DEFT retains more authentic anatomical features and textures, reflecting a more realistic depiction of the animals. On the other hand, LoRA brings a more artistic flair to the HORSE images, with creative use of colors, dynamic poses, and abstract elements. While LoRA's outputs are more vibrant and imaginative, they may not always preserve the natural look and feel of the animals as consistently as DEFT does.

Despite LoRA's higher scores in DINOv1 and DINOv2, these results do not fully capture its ability to maintain realistic features across all images. LoRA excels in producing creative, artistic representations, but at the cost of some consistency in realism. DEFT, with its emphasis on realism, demonstrates more stable, high-fidelity outputs, particularly for complex subjects like horses.

This analysis shows that while LoRA excels in artistic flexibility and creative interpretation, achieving higher DINOv1 and DINOv2 scores, DEFT remains superior in generating more realistic and detailed images. The choice between the two methods ultimately depends on the desired outcome—whether the goal is to prioritize artistic creativity or to maintain realistic accuracy.

### S1.0.4 Scene personalization

Figures 11 and 12 showcase the model's scene personalization capabilities, demonstrating its proficiency in generating high-quality visual content with specific environmental characteristics.

| Method | Image | DINOv1 | DINOv2 | CLIP-I | CLIP-T |
|---|---|---|---|---|---|
| DEFT | Kitten | 79.9416 | 73.5553 | 83.5867 | 35.5409 |
| DEFT | Stork | 67.9085 | 62.7926 | 76.3276 | 36.8799 |
| DEFT | Kitten 2 | 77.8501 | 65.0358 | 77.6003 | 36.7579 |
| DEFT | HORSE | 77.8501 | 65.0358 | 77.6003 | 36.7579 |
| LoRA | Kitten | 82.9492 | 77.9071 | 86.7823 | 33.7977 |
| LoRA | Stork | 70.1220 | 61.8301 | 74.9231 | 36.9636 |
| LoRA | Kitten 2 | 83.5538 | 72.2653 | 81.3446 | 34.1923 |
| LoRA | HORSE | 83.5538 | 72.2653 | 81.3446 | 34.1923 |

Table 8: Comparison of DEFT and LoRA across multiple evaluation metrics (DINOv1, DINOv2, CLIP-I, CLIP-T) for images of cats and horses. The table highlights how LoRA consistently achieves higher scores in DINOv1 and DINOv2, indicating its strength in capturing complex features, while DEFT excels in CLIP-I and CLIP-T, reflecting its focus on realism and detailed preservation of the original subjects.

Figure 11, titled Church Rock Scene Personalization, illustrates how the model adapts the Church Rock scene to various settings. These scenes include dynamic backgrounds, such as a pool surrounded by palm trees, a futuristic city at night, a snowy mountain top, a crowded street market, and a forest with autumn leaves. Each personalized scene is a result of fine-tuning, reflecting how the model can generate diverse visual representations of the same object in unique environments, showcasing its flexibility in handling scene-specific details.

Figure 12, titled Table Scene Personalization, further exemplifies the model's ability to adapt to specific environments. In this case, the model personalizes a simple table scene by generating various configurations of objects like bottles and caps, based on detailed prompts. The generated scenes show different bottles, one filled with orange liquid and another empty, both with distinct cap colors. This reflects the model's ability to generate high-quality content by adapting to specific setups, maintaining both object clarity and spatial coherence within the scene.

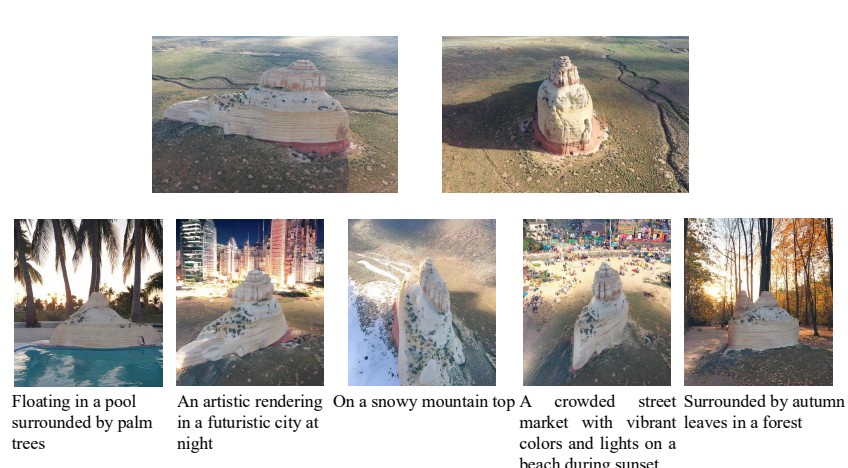

Floating in a pool surrounded by palm trees • An artistic rendering in a futuristic city at night • On a snowy mountain top • A crowded street market with vibrant colors and lights on a beach during sunset • Surrounded by autumn leaves in a forest

Figure 11: This figure illustrates the scene personalization capabilities of the model using the church rock scene, showcasing how fine-tuning allows for detailed control over scene-specific characteristics.

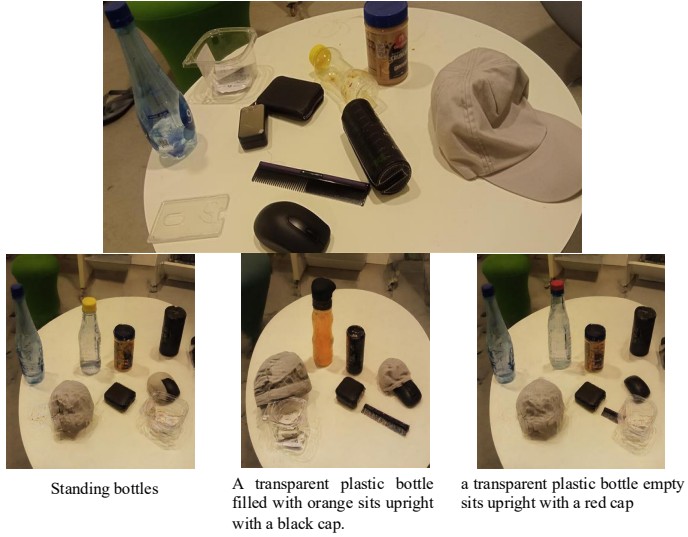

| Standing bottles | A transparent plastic bottle filled with orange sits upright with a black cap. | a transparent plastic bottle empty sits upright with a red cap |

Figure 12: Table Scene Personalization: The figure demonstrates how the model personalizes a table scene, reflecting the ability to adapt and generate high-quality visual content in specific environments.

