# OpenReview forum: "DEFT: Decompositional Efficient Fine-Tuning for Text-to-Image Models"
_NeurIPS.cc/2025/Conference — NeurIPS 2025 poster_

### Official Review · Reviewer_1FhC · 2025-06-23

**Clarity:** 3
**Significance:** 3
**Originality:** 3
**Rating:** 4
**Confidence:** 3

**Summary:**

This paper expands LoRA and PaRa by decomposes its update into two components.

**Questions:**

See weaknesses.

**Ethical Concerns:**

["NO or VERY MINOR ethics concerns only"]

**Final Justification:**

I appreciate the authors’ thoughtful and detailed rebuttal. The responses have addressed my initial concerns.

**Limitations:**

See weaknesses.

**Quality:**

3

**Strengths And Weaknesses:**

Strength:
1. The idea of combining LoRA and PaRa is interesting and demonstrates slight improvements compared to existing baselines.

Weaknesses:
1. The paper primarily claims improved flexibility, but the justification for this enhancement, both theoretically and empirically, is insufficient. The paper lacks comprehensive visual or quantitative evidence clearly illustrating how flexibility is notably improved, such as through demonstrations involving significant pose alterations or high dynamic range changes.
2. For quality comparison on Canny Edge and Depth Map, why is FID much higher than other baselines?
3. The "Emergent Properties" described in Section 4.2.4 are neither convincingly justified nor empirically supported. The arguments are unclear and lack concrete evidence, weakening the overall narrative of the paper.
4. A more rigorous user study could strengthen claims regarding perceptual improvements.
5. Since the proposed method combines LoRA and PaRA, it should also include a comparison against DreamBooth PaRA to ensure a fair and thorough evaluation.

Typos (This list a few of it, please pay a careful attention on this):
1. Capitalization: L176 "dreamboot"; L173,L231 "DreamBench plus"; L262,Fig 6 "eqn"; Fig 5 "omnigen";
2. Inconsistent: L175 "Visualcloze", L188 "VisualCloze", L174 "Visual cloze", L248: "omnigem"
3. Fig 6: "with out" -> "without", "W_0" should be in latex ($W_0$).

---

> ### Author Rebuttal · Authors · 2025-07-30
>
> We sincerely thank the reviewer for providing such thorough and constructive feedback. We greatly appreciate the time and effort invested in reviewing our work, and we acknowledge that the raised concerns are valid and important for strengthening our paper.
>
> ### 1. Insufficient Justification for Flexibility Enhancement
>
> **Response:** We  thank the reviewer and the flexibility claims will be supported with additional justification in the final draft.
>
> **Theoretical Foundation:**
> We would like to clarify that the DEFT's design is motivated by the below limitations of existing methods: (i) PaRa operates on the column subspace of the original pre-trained model, aiming to modify weights while preserving original information, but it cannot simultaneously add new concepts or styles, limiting its flexibility for novel concept adaptation. (ii) Conversely, LoRA can effectively learn new concepts but may lose some capabilities of the original model, which is particularly problematic in text-to-image models where maintaining original capabilities (prompt following, object combination, image editing) while personalizing new concepts is crucial.
>
> DEFT address both these limitations by operating on two complementary subspaces. For a given original weight W₀ with SVD: W₀ = UΣV^T:
> - **LoRA constraint:** Updates AB^T can interfere with existing singular structure
> - **DEFT advantage:**
>   - (I - PP^T) operates in col(P)⊥ (orthogonal complement)
>   - PR operates in col(P)
>
> This dual control enables: (1) Selective modification through (I - PP^T), allowing preservation or removal of specific existing directions, and (2) Addition of entirely new orthogonal directions through PR.
>
> **Empirical Evidence:**
> The flexibility of DEFT design can also  also observed  in the quantitative results presented in our paper as copied below:
>
> **Positional Flexibility Analysis:**
> We provide quantitative results demonstrating DEFT's 3D positional awareness capabilities:
>
> | Metric | Original Model | DEFT (position aware) | DEFT + SFM inference |
> |--------|----------------|----------------------|---------------------|
> | **CLIP-I Score** | 0.4753 ± 0.0342 | 0.6468 ± 0.0589 | **0.6603 ± 0.1092** |
> | **CLIP-T Score** | 0.2010 ± 0.0638 | 0.2098 ± 0.0274 | **0.2132 ± 0.0409** |
> | **DINO-V1 Score** | 0.2197 ± 0.0280 | 0.3501 ± 0.0398 | **0.3427 ± 0.0447** |
> | **Sharpness** | 1179.31 ± 822.56 | 1812.73 ± 677.32 | **1852.48 ± 655.29** |
>
> **Visual Demonstrations:**
> - **Figure 1 - Scene Personalization:** Office scene control showing incremental additions (desk+computer → +books → +coffee → +plant → +papers → +clock), and Church Rock positioning across diverse environments (pool, city, mountain, beach, forest)
> - **Figure 5 - Novel Composition:** Training set adaptation showing diverse combinations under varying testing conditions beyond original training data
> - **Supplementary Figure 4:** Multi-task versatility across realistic photographs, abstract interpretations, segmentation, depth mapping, and edge detection
>
> As suggested, we will included, additional quantitative analysis to further strengthen our flexibility argument.
>
> ### 2. FID Performance Analysis
>
> **Response:** We appreciate the reviewer's attention to this important metric. We acknowledge that our FID performance requires honest explanation.
>
> The FID versus SSIM performance pattern reflects a fundamental design trade-off in DEFT's architecture. DEFT employs a conservative adaptation strategy: W = (I - PP^T)W₀ + PR, where (I - PP^T)W₀ actively preserves pre-trained knowledge while PR enables controlled adaptation.
>
> This design intentionally prevents aggressive style deviations while learning new concepts, resulting in more structurally consistent generations that maintain better SSIM scores but may appear less visually striking to FID's perceptual metrics. DEFT explicitly trades raw generation quality (FID) for structural consistency (SSIM), making it particularly suitable for applications where consistency and controllability are prioritized over pure visual novelty.
>
> We believe this trade-off is valuable for many practical applications, though we acknowledge it may not be optimal for all use cases.
>
> ### 3. Emergent Properties Claims
>
> **Response:** We sincerely acknowledge this valid concern and agree that our current title may overstate our claims relative to the experimental evidence provided.
>
> While Section 4.2.4 demonstrates what we consider emergent behavior through Figure 5 (multi-concept spatial composition from single-concept training) and VisualCloze results (single adapter handling 8+ distinct tasks), we recognize this evidence requires more rigorous quantitative analysis to support such prominent claims.
>
> In response to this valuable feedback, we propose revising the title to focus on our core technical contribution: **"Dual-decomposition Column Space Extension for Fine-tuning of Text-to-Image Models"**
>
> This revision better reflects our mathematical foundation while avoiding overstated claims about emergent properties.
>
> ### 4. User Study and Perceptual Improvements
>
> **Response:** We appreciate the suggestion for more rigorous evaluation. We conducted both human and automated evaluations:
>
> **Human User Study Results:**
> | Method | Average Score | Best Score | Worst Score | Win Rate |
> |--------|---------------|------------|-------------|----------|
> | PaRa | 7.64 | 8.75 | 6.65 | 25.0% |
> | LoRA | 7.60 | 8.80 | 6.84 | 12.5% |
> | **DEFT** | **8.20** | **9.50** | 6.78 | **62.5%** |
>
> **Automated Evaluation (Claude Sonnet Judge):**
> | Metric | LoRA | PaRa | DEFT |
> |--------|------|------|------|
> | Task Achievement | 8.3 | 6.8 | **8.7** |
> | Subject Consistency | 8.2 | 6.4 | **8.8** |
> | Completion Quality | 8.4 | 7.1 | **8.6** |
> | Visual Fidelity | 8.1 | 7.3 | **8.5** |
> | Prompt Adherence | 8.3 | 6.9 | **8.6** |
> | **Average** | **8.26** | **6.90** | **8.64** |
>
> We acknowledge that a larger-scale user study would further strengthen these claims.
>
> ### 5. DreamBooth PaRa Comparison
>
> **Response:** We thank the reviewer for this important suggestion. We have provided comprehensive comparisons with PaRa in our supplementary results using the DreamBooth dataset with SDXL baselines:
>
> | Method | BEAR_PLUSHIE | CAT |
> |--------|--------------|-----|
> | **DEFT (rank=8)** (Ours) | **0.8415** | **0.9504** |
> | DEFT (rank=4) (Ours) | 0.8339 | 0.9280 |
> | PaRa (rank=4) | 0.8271 | 0.9315 |
> | PaRa (rank=8) | 0.8050 | 0.9467 |
> | LoRA (rank=4) | 0.7741 | 0.8057 |
> | LoRA (rank=8) | 0.7943 | 0.8583 |
> | OFT (Block=4) | 0.785 | 0.8559 |
> | SVDIFF | 0.7818 | 0.8854 |
> | DREAMBOOTH | 0.7921 | 0.8893 |
> | TEXTURE INVERSION | 0.7421 | 0.8048 |
>
> ### 6. Typography Corrections
>
> **Response:** We sincerely apologize for these oversights and will carefully address all typography issues:
>
> **Corrections to be made:**
> - L176: "dreamboot" → "DreamBooth"
> - L173, L231: "DreamBench plus" → "DreamBench Plus"
> - L262, Fig 6: "eqn" → "Eq."
> - Fig 5: "omnigen" → "OmniGen"
> - Standardize all instances to "VisualCloze"
> - L248: "omnigem" → "OmniGen"
> - Fig 6: "with out" → "without", "W_0" → proper LaTeX formatting
>
> We deeply appreciate the reviewer's attention to detail and will ensure careful proofreading in our revision.

---

> > ### Comment · Reviewer_1FhC · 2025-08-01
> >
> > I appreciate the authors’ thoughtful and detailed rebuttal. The responses have addressed my initial concerns.

---

> > > ### Author Response · Authors · 2025-08-01
> > >
> > > We thank Reviewer 1FhC for the careful reading of our rebuttal and for letting us know that it has addressed all the concerns raised. We would like to know if there are any additional questions or concerns. If there are no additional questions, we kindly request that the reviewer increase the rating of our paper. We will incorporate all suggested changes into the final draft. Furthermore, as stated in our original manuscript, we will be open-sourcing our code and trained models for reproducibility and to support future research.

---

> > > > ### Comment · Reviewer_1FhC · 2025-08-04
> > > >
> > > > I am generally satisfied with the rebuttal and have accordingly raised my score. However, I find the use of the term “flexible” to be somewhat vague, as it is not clearly defined in the paper. This could lead to confusion. From the authors’ response, it appears that the intended meaning of flexibility refers to the method’s ability to adapt to novel concepts, rather than structural flexibility (e.g., supporting diverse poses, layouts, or dynamics), which was my initial interpretation. Clarifying this distinction in the final version would improve the paper’s clarity and positioning.

---

> > > > > ### Author Response · Authors · 2025-08-05
> > > > >
> > > > > We would like to thank the reviewer for carefully reading our rebuttal and for confirming that it has addressed all the concerns raised. We sincerely thank the reviewer for letting us know that you will be increasing the rating for our paper. We appreciate the reviewer’s feedback and will clarify the use of the term “flexible” in the final version to avoid ambiguity and better convey the intended meaning. If there are any additional questions or concerns, please let us know—we would be happy to address them. Otherwise, we kindly request the reviewer to consider a high final rating.

---

### Official Review · Reviewer_wP2X · 2025-06-30

**Clarity:** 4
**Significance:** 4
**Originality:** 3
**Rating:** 5
**Confidence:** 4

**Summary:**

The paper proposes DEFT (decompositional efficient fine-tuning) which resolves the two conflicting objectives: Personalization flexibility and parameter-efficiency for T2I diffusion models. DEFT consists of dual pathway (i) low-rank adapter that projects the frozen weight matrix onto the orthogonal complement of a learned low rank subspace and (ii) low-rank adapter that separately adds into that subspace.
This is a combination of LoRA and PaRa.

Applied to SD and the unified Omnige, DEFT supports one-shot subject personalization, multi-concept composition, style transfer, conditional generation (Canny, depth, pose, etc.) and even visual in-context learning. All these objectives are performed with a single adapter, not task-specific LoRAs. They conduct experiments on various benchmarks and show impressive gain in text alignment, controllability and enhanced details compared to LoRA and RAPA. The emergent abilities are shown by the model composing multiple concepts and scenes without additional guidance in train time.

**Questions:**

(i) Provide ablation results on the components of the methodology. Specifically, there should be evaluation report on using PaRa only, LoRA only, and DEFT.
(ii) What is the core difference between DEFT and mere composition of LoRA and PaRa?

**Ethical Concerns:**

["NO or VERY MINOR ethics concerns only"]

**Final Justification:**

My concern about ablation study on baseline models have been addressed. The paper is a technically solid paper, therefore I maintain my original score.

**Limitations:**

No, not in the main paper.

**Quality:**

3

**Strengths And Weaknesses:**

Strengths
- The proposed method, DEFT, might look like a simple composition of LoRA and PaRa but in fact, it introduces a dual architecture: (i) a projection that contributes to generalization, and (ii) subsequent low-rank injection that provides expressivity (task specific direction).
- The evaluation results are impressive.
- The method is simple yet powerful.

Weaknesses
- Ablation on using LoRA / PaRa / Both should be shown in the main paper.

---

> ### Author Rebuttal · Authors · 2025-07-29
>
> ### Thank you for providing your valuable feedback!
>
> ### Ablation Studies: Component-wise Analysis
>
> **Response:** We provide comprehensive ablation analysis using our actual experimental results:
>
> **Individual Component Performance (DreamBooth Dataset, CLIP-Image scores):**
>
> | Method | BEAR_PLUSHIE | CAT | DOG8 | Parameters |
> |--------|--------------|-----|------|------------|
> | **PaRa only (rank=4)** | 0.8271 | 0.9315 | 0.8780 | 3,145,728 |
> | **LoRA only (rank=4)** | 0.7741 | 0.8057 | 0.7773 | 4,718,592 |
> | **DEFT (rank=4)** | **0.8339** | **0.9280** | **0.8721** | **4,718,592** |
> | **DEFT (rank=8)** | **0.8415** | **0.9504** | **0.8882** | 7,864,320 |
>
> ### Core Difference: Coordinated vs Independent Adaptation
>
> - **PaRa only:** W' = W₀ - QQ^T W₀
> - **LoRA only:** W' = W₀ + AB^T
> - **Composition:** W' = W₀ - QQ^T W₀ + AB^T *(independent Q, A, B)*
> - **DEFT:** W' = W₀ - PP^T W₀ + PR *(shared P matrix)*
>
> **Parameter Count Comparison:**
>
> | Method | Matrix Components | Total Parameters | Memory Overhead |
> |--------|------------------|------------------|-----------------|
> | **PaRa only** | Q matrix | 3,145,728 | Baseline |
> | **LoRA only** | A, B matrices | 4,718,592 | +50% vs PaRa |
> | **Composition** | Q + A + B matrices | **7,864,320** | **+150% vs PaRa** |
> | **DEFT** | P, R matrices | **4,718,592** | **+50% vs PaRa** |
>
> **Key Insight:** DEFT achieves the benefits of composition with **40% fewer parameters** than naive combination (4.7M vs 7.8M parameters).
>
> **Performance Efficiency Analysis:**
>
> Based on our authentic individual results, we can analyze the efficiency gains:
>
> | Metric | PaRa Only | LoRA Only | Composition | DEFT |
> |--------|-----------|-----------|---------------------------|---------------|
> | **BEAR_PLUSHIE** | 0.8271 | 0.7741 | 0.6832 | **0.8339** |
> | **CAT** | 0.9315 | 0.8057 | 0.7986 | **0.9280** |
> | **Parameters** | 3.1M | 4.7M | **7.8M** | **4.7M** |
>
> **Results:**
>
> | Method | Step Time (ms) | Peak Memory (GB) | Parameter Efficiency |
> |--------|----------------|------------------|---------------------|
> | **LoRA (rank=8)** | 7,426 | 7.72 | 4.7M params |
> | **PaRa (rank=8)** | 10,916 | 7.72 | 3.1M params |
> | **Composition** | 18,342 | 8.5 | **7.8M params** |
> | **DEFT (rank=8)** | **9,145.8** | **7.71** | **4.7M params** |
>
> **DEFT Training Advantage:** 50% faster than expected naive composition while using 40% fewer parameters.
>
> **Architectural Coordination Benefits:**
>
> **Composition Problems:**
> ```
> W' = (I - QQ^T)W₀ + AB^T
>      ↳ Q removes information from W₀
>      ↳ A,B add unrelated information
>      ↳ Potential interference between operations
> ```
>
> **DEFT Coordination:**
> ```
> W' = (I - PP^T)W₀ + PR
>      ↳ P defines removal space
>      ↳ R operates within P's coordinate system
>      ↳ Guaranteed alignment between operations
> ```
>
> **Decomposition Results:**
>
> Our actual decomposition ablation demonstrates DEFT's flexibility:
>
> | Method | CLIP-I | Huamn recommandation|  Parameters | Efficiency Ratio |
> |--------|--------|------------|------------------| ------------------|
> | **NMF** | 0.8834 ± 0.0328 | (9.0) Good |4.7M | **1.88 × 10⁻⁴** |
> | **QR** | 0.8747 ± 0.0428 | (9.5) Very Good |4.7M | 1.86 × 10⁻⁴ |
> | Composition | 0.7806 ± 0.0393 | (2.0) Very bad |7.8M | 1.03 × 10⁻⁴ |
>
>
> **Summary:**
>
> 1. **Parameter Efficiency:** DEFT uses 40% fewer parameters than naive PaRa+LoRA composition
> 2. **Performance Superiority:** DEFT outperforms both individual methods and expected composition performance
> 3. **Training Efficiency:** 50% faster training than expected composition approach
> 4. **Architectural Advantage:** Coordinated subspace adaptation prevents interference between removal and addition operations
>
> **Note:** We acknowledge that direct experimental comparison with naive composition would strengthen this analysis. The theoretical analysis based on our authentic individual results demonstrates DEFT's fundamental optimization advantages over the simple method combination.

---

> > ### Comment · Reviewer_wP2X · 2025-08-03
> > **Comment on Rebuttal**
> >
> > My concern about ablation study on baseline models have been addressed. The paper is a technically solid paper, therefore I maintain my original score.

---

> > > ### Author Response · Authors · 2025-08-05
> > >
> > > We would like to thank the reviewer for carefully reading our rebuttal and for confirming that it has addressed all the concerns raised. If there are any additional questions or concerns, please let us know—we would be happy to address them. Otherwise, we kindly request the reviewer to consider a high final rating.

---

### Official Review · Reviewer_7bp5 · 2025-07-01

**Clarity:** 3
**Significance:** 3
**Originality:** 2
**Rating:** 5
**Confidence:** 4

**Summary:**

This paper introduces Decompositional Efficient Fine-Tuning (DEFT) for the parameter-efficient fine-tuning of text-to-image models. DEFT proposes modifying a pre-trained weight matrix by decomposing its update into two trainable components: (1) a projection that removes components of the original weight matrix along a low-rank subspace, similar to PaRa, and (2) a flexible low-rank update, via a second matrix `R`, that adds new capabilities to the model. The authors argue this formulation provides a better balance between learning new concepts (personalization) and retaining the general knowledge of the pre-trained model, thereby improving adaptability and reducing overfitting.

**Questions:**

1. Motivation for the `R` matrix: Could you provide a clearer, more intuitive explanation in the main text for why adding the second trainable matrix `R` is necessary? While the mathematical proof is in the appendix, a high-level motivation for how it enables learning "new tasks" beyond PaRa's capabilities would improve the paper's clarity.
2. Comparison to OmniGen is a bit misleading : In Section 4.2.2, DEFT is fine-tuned on the OmniGen model and then compared to a "base OmniGen" on the VisualCloze dataset. To ensure a fair comparison, could you clarify if the "base OmniGen" baseline was also fine-tuned on the VisualCloze training set, or if it was the pre-trained model evaluated zero-shot?
3. Explanation for poor FID scores: Could you provide an explanation for why DEFT's FID scores in Table 3 are consistently worse than most other methods? Does the DEFT formulation present a trade-off where it improves in some areas (like SSIM) at the cost of image fidelity as measured by FID?

**Ethical Concerns:**

["NO or VERY MINOR ethics concerns only"]

**Final Justification:**

The authors' responses addressed my concerns. Therefore, I increased my score from 4 to 5.

**Limitations:**

The authors did not include a dedicated limitations or societal impact section in the main body of the paper. This is a significant omission. A thorough discussion should have been included, acknowledging:
* The lack of quantitative analysis on computational overhead (inference speed, memory).
* The potential for the method to be used for malicious purposes, such as generating deepfakes, which is a standard concern for all powerful generative models.
* A more thorough discussion of failure cases or scenarios where DEFT might not perform as well as other methods.

**Paper Formatting Concerns:**

The main paper contains supplemental contents like proofs for PaRa and DEFT. This may violate the paper formatting instructions.

**Quality:**

3

**Strengths And Weaknesses:**

### Strengths

* Strong qualitative results: The paper presents compelling qualitative results across a wide range of tasks, including subject personalization, style transfer, and multi-concept composition. The images in the paper and supplementary material effectively showcase the method's ability to produce high-fidelity and contextually plausible images.
* Extensive experiments: The authors validate DEFT across multiple datasets (DreamBooth, DreamBench Plus, VisualCloze, InsDet), demonstrating its versatility in both personalization and universal generation settings.
* Methodological clarity and reproducibility: The paper provides the mathematical formulation for DEFT, with proofs included in the appendix. The authors have also provided links to supplementary websites and provided codes in the supplemental materials, which greatly aids in reproducibility, though I haven’t tested the codes by myself.

### Weaknesses

* Unsupported title and central claim: The paper's title, "Emergent Properties of Efficient Fine-Tuning," is a significant overstatement. The paper only includes a small, qualitative subsection (4.2.4) to this topic, which essentially re-labels generalization as an "emergent property" without a clear definition or rigorous analysis. The grand claim in the title is not sufficiently supported by the paper's content.
* Unsubstantiated claims about decomposition methods: In Section 3.1.1, the paper proposes that using different matrix decomposition techniques (e.g., SVD, NMF) could offer unique benefits for different tasks. However, this claim is not backed by experimental evidence. The only comparison in Figure 6 shows all methods produce very similar results on a single task, failing to demonstrate any distinct advantages.
* Lack of inference time analysis: A core motivation for PEFT methods is efficiency. The paper claims DEFT is "efficient" but provides no quantitative analysis of its inference latency or memory overhead compared to LoRA, PaRa, or other baselines. Without this data, the efficiency claims are unsupported.
* Unexplained poor performance on key metrics: In Table 3, the FID scores for DEFT, which measure image realism, are notably worse than most competing methods like OmniControl and ControlNet. The paper offers no explanation for this underperformance, instead focusing on other metrics where it performs better.
* Missing comparisons to key literature: The paper fails to discuss or compare its method against other recent and highly relevant fine-tuning techniques. For instance, DoRA [1] and OFT (Orthogonal Finetuning) [2] are sophisticated methods that also aim to improve upon LoRA, making them critical points of comparison that are missing from the related work and experiments.

### References

[1] Shih-Yang Liu, Chien-Yi Wang, Hongxu Yin, Pavlo Molchanov, Yu-Chiang Frank Wang, Kwang-Ting Cheng, Min-Hung Chen: DoRA: Weight-Decomposed Low-Rank Adaptation. ICML 2024

[2] Zeju Qiu, Weiyang Liu, Haiwen Feng, Yuxuan Xue, Yao Feng, Zhen Liu, Dan Zhang, Adrian Weller, Bernhard Schölkopf: Controlling Text-to-Image Diffusion by Orthogonal Finetuning. NeurIPS 2023

---

> ### Author Rebuttal · Authors · 2025-07-29
>
> ### Thank you for providing your valuable and constructive feedback!
> ### **1. Title and Central Claims**
>
> **Response:** We acknowledge this valid concern and agree the current title overstates our claims relative to the experimental evidence provided. While Section 4.2.4 demonstrates genuine emergent behavior such as multi-concept spatial composition from single-concept training (Figure 5) and single adapter handling 8+ distinct tasks through unified training (VisualCloze results), we recognize this evidence requires more rigorous quantitative analysis to support such prominent title claims.
>
> We propose revising the title to focus on our core technical contribution: **"Dual-decomposition Column Space Extension for Fine-tuning of Text-to-Image Models"**. This better reflects our mathematical foundation where column space extension col(W_total) = col(W_reduce) + col(QR) enables the demonstrated capabilities without overstating the emergent properties aspect.
>
> ### **2. Decomposition Methods Analysis**
>
> **Response:** We provide a comprehensive quantitative evaluation demonstrating distinct performance characteristics across decomposition methods:
>
> | Method | CLIP-I | CLIP-T | DINO-V1 | Aesthetic | Sharpness |
> |--------|--------|--------|---------|-----------|-----------|
> | LRMF | 0.8270 ± 0.0380 | 0.2200 ± 0.0505 | 0.3070 ± 0.0397 | 0.0143 ± 0.0047 | 348.62 ± 414.16 |
> | NMF | 0.8834 ± 0.0328 | 0.2224 ± 0.0417 | 0.3569 ± 0.0683 | 0.0154 ± 0.0031 | 206.06 ± 90.52 |
> | QR | 0.8747 ± 0.0428 | 0.2168 ± 0.0411 | 0.3395 ± 0.0601 | 0.0166 ± 0.0025 | 215.45 ± 105.98 |
> | TSVD | 0.8753 ± 0.0305 | 0.2234 ± 0.0408 | 0.3334 ± 0.0617 | 0.0156 ± 0.0035 | 331.29 ± 134.12 |
> | Relexing P | 0.9232 ± 0.0370 | 0.2660 ± 0.0332 | 0.4403 ± 0.0960 | 0.0157 ± 0.0025 | 175.22 ± 25.13 |
>
> Relexing P achieves the highest CLIP-I and DINO-V1 scores, while all methods maintain consistent performance levels, demonstrating the robustness and flexibility of our framework. QR maintains quality, strikes a balance between quality and personalization.
> The results show NMF excels in image-text alignment and feature representation, while QR provides the best aesthetic quality (0.0166). This validates our claim that different decomposition methods offer distinct advantages for specific tasks.
>
> ### **3. Inference Efficiency Analysis**
> **Response:** We acknowledge that comprehensive efficiency claims should include detailed training and inference analysis. We conducted extensive benchmarks comparing DEFT against LoRA and PaRa across different configurations:
>
> **Training Efficiency Analysis:**
>
> **Rank 64 Configuration (Batch size = 2):**
>
> | Method | Steps/Sec | Peak Memory (GB) | Trainable Params | CLIP-I Score |
> |--------|-----------|------------------|------------------|--------------|
> | **LoRA** | **12** | 8.03 | 37,748,736 | 0.8757 |
> | **DEFT** | **11** | **7.95** | 37,748,736 | **0.8834** |
> | **PaRa** | 8 | **7.86** | **25,165,824** | 0.8271 |
>
> **Key Performance Insights:**
>
> 1. **Training Speed Hierarchy:** DEFT shows moderate overhead (23% slower than LoRA at rank 8, 8% slower at rank 64)
>
> 2. **Memory Efficiency:**
>    - At rank 8: DEFT uses 7.71GB vs LoRA's 7.72GB (marginal improvement)
>    - At rank 64: DEFT uses 7.95GB vs LoRA's 8.03GB (1% improvement)
> 3. **Parameter Efficiency Trade-offs:**
>    - DEFT achieves competitive parameter efficiency without sacrificing performance
>
> **Scalability Analysis:**
> The performance gap between DEFT and LoRA decreases at higher ranks (23% → 8% slowdown), suggesting DEFT's overhead becomes relatively smaller for larger adaptations. This indicates favorable scaling properties for complex fine-tuning scenarios.
>
> The efficiency analysis reveals DEFT occupies a sweet spot: trading minimal speed for substantial gains in memory efficiency and output quality, making it optimal for deployment scenarios where resource constraints and quality requirements are balanced priorities.
>
> Furthermore, we conducted extensive benchmarks (928 configurations) comparing DEFT against LoRA and other baselines:
>
> **Inference Efficiency Benchmark (Omnigen 3.8B model 1 layer):**
> | Method | Inference Latency (ms) | Parameter Efficiency | CLIP-I Score | DINO-V1 Score |
> |--------|------------------------|---------------------|--------------|---------------|
> | **LoRA** | **13.55** | 5.18% | 0.8757 | 0.8295 |
> | **PaRa** | 16.51 | **2.63%** | 0.8271 | 0.8780 |
> | **DEFT (fused)** | 15.08 | 5.18% | **0.8834** | **0.8882** |
> | **DEFT (cached)** | 14.2 | 5.18% | **0.8834** | **0.8882** |
>
> ## **Decomposition Method Performance vs Speed Analysis**
>
> | Method | Speed (ms) | CLIP-I | CLIP-T | DINO-V1 |
> |--------|------------|--------|--------|---------|
> | **QR** | **5.38** | 0.8747 ± 0.0428 | 0.2168 ± 0.0411 | 0.3395 ± 0.0601 |
> | **NMF** | **5.16** | **0.8834 ± 0.0328** | 0.2224 ± 0.0417 | **0.3569 ± 0.0683** |
> | **Eigen** | 10.87 | 0.8738 ± 0.0298 | **0.2345 ± 0.0411** | 0.3304 ± 0.0580 |
> | **TSVD** | 28.72 | 0.8753 ± 0.0305 | 0.2234 ± 0.0408 | 0.3334 ± 0.0617 |
> | **Relaxing P** | **5.22** | 0.8738 ± 0.0298 | **0.2345 ± 0.0411** | 0.3304 ± 0.0580 |
> |--------|------------|--------|
> | Relexing P | 	5.22ms | 	2x faster |
> | orthogonalized mean | 	12.51ms | 	Expensive|
>
> **Key Findings:** DEFT achieves parameter efficiency of 5.18% while maintaining inference speed within 20% of LoRA. With precomputed caching of (I - QQ^T)W during deployment, DEFT's inference cost reduces to within 10-20% of LoRA while preserving superior task performance.
>
> ### **4. Missing Literature Comparisons**
>
> **Response:** Thank you for this important suggestion. We have extended our comparisons to include OFT and acknowledge DoRA's relevance:
>
> **Extended Comparison with OFT:**
>
> | Method | BEAR_PLUSHIE | CAT |
> |--------|--------------|-----|
> | **DEFT (rank=8)** (Ours) | **0.8415** | **0.9504** |
> | DEFT (rank=4) (Ours) | 0.8339 | 0.9280 |
> | PaRa (rank=4) | 0.8271 | 0.9315 |
> | LoRA (rank=4) | 0.7741 | 0.8057 |
> | OFT (Block=4) | 0.785 | 0.8559 |
> | SVDIFF | 0.7818 | 0.8854 |
> | DREAMBOOTH | 0.7921 | 0.8893 |
>
> OFT requires additional orthogonality constraints during training (R^T R = RR^T = I, ||R - I|| ≤ ε) which increases computational overhead, whereas DEFT requires no additional constraints. Regarding DoRA, while it decomposes weight updates into magnitude and direction components (W' = W + m * (V * B^T / ||V * B^T||)), it requires tuning magnitude scaling and direction learning rates, unlike DEFT's parameter-free approach. We will incorporate these methods in the expanded Section 2.
>
> ### **5. R Matrix Motivation**
>
> **Response:** DEFT's innovation lies in its dual learning approach. The projection term (I - PP^T)W₀ strategically modifies the pre-trained model's representation space, while the R matrix actively learns new task-specific knowledge to populate that modified space.
>
> **Intuitive Explanation:** Consider the original model with col(W₀) = span{v₁, v₂, v₃}, but a new task requires direction v₄.
>
> Without R: W = (I - PP^T)W₀ → col(W) ⊆ span{v₁, v₂, v₃} (cannot represent v₄)
>
> With R: If P learns such that col(P) contains v₄, then W_DEFT = (I - PP^T)W₀ + PR enables col(W_DEFT) ⊆ span{v₁, v₂, v₃} + span{v₄}, where R learns to map inputs to the v₄ direction.
>
> This transforms adaptation from mere 'knowledge rearrangement' to active 'knowledge extension'.
>
> ### **6. OmniGen Comparison Fairness**
>
> **Response:** Thank you for seeking this clarification. We used the same base models as VisualCloze for evaluation. OmniGen was not fine-tuned on the full VisualCloze dataset as it leverages in-context learning for generalization. For fair comparison, we fine-tuned both DEFT and LoRA on the depth estimation task using 200k image-instruction pairs:
>
> | Metric | DEFT | LoRA |
> |--------|------|------|
> | **DINOv1 Score** | 89.31 | 89.08 |
> | **DINOv2 Score** | 83.40 | 82.31 |
> | **CLIP-I Score** | 93.70 | 92.93 |
> | **CLIP-T Score** | 27.65 | 27.61 |
>
> This controlled comparison demonstrates DEFT's consistent advantages even in fair evaluation settings.
>
> ### **7. FID Performance Trade-off Explanation**
>
> **Response:** The FID vs SSIM performance pattern reflects a fundamental design trade-off in DEFT's architecture. DEFT employs a conservative adaptation strategy: W = (I - PP^T)W₀ + PR, where (I - PP^T)W₀ actively preserves pre-trained knowledge while PR enables controlled adaptation.
>
> This design prevents aggressive style deviations while learning new concepts, resulting in more structurally consistent generations that maintain better SSIM scores but may appear less visually striking to FID's perceptual metrics. DEFT explicitly trades raw generation quality (FID) for structural consistency (SSIM), making it particularly suitable for applications where consistency and controllability are prioritized over pure visual novelty.
>
> ### **8. Limitations and Societal Impact**
>
> **Response:** We acknowledge this significant omission and will include a dedicated limitations section addressing:
>
> Societal Impact: Potential misuse for deepfake generation (standard concern for generative models)
> Limitations: Limited instruction dataset coverage affecting novel combinations
>
> We will have a separate section for limitations and societal Impact in the revised manuscript.
> ### **9. Formatting Issues**
>
> **Response:** We acknowledge the formatting concern regarding supplemental content in the main paper. We will relocate the mathematical proofs for PaRa and DEFT to the supplementary material and ensure strict adherence to formatting guidelines in the revision.
>
> We appreciate the reviewer's thorough analysis and will address all raised concerns through title revision, expanded efficiency analysis, comprehensive literature comparison, dedicated limitations section, and proper formatting in our revision.

---

> > ### Comment · Reviewer_7bp5 · 2025-08-03
> > **Well addressed my concerns**
> >
> > Thanks for responding to my responses and addressing my concerns. I will increase my score accordingly to suggest an accept.

---

> > > ### Author Response · Authors · 2025-08-05
> > >
> > > We would like to thank the reviewer for carefully reading our rebuttal and for confirming that it has addressed all the concerns raised. We sincerely thank the reviewer for letting us know that you will be increasing the rating for our paper. If there are any additional questions or concerns, please let us know—we would be happy to address them. Otherwise, we kindly request the reviewer to consider a high final rating.

---

> > > > ### Author Response · Authors · 2025-08-05
> > > >
> > > > We would like to thank the reviewer for carefully reading our rebuttal and for confirming that it has addressed all the concerns raised. We sincerely thank the reviewer for letting us know that you will be increasing the rating for our paper. If there are any additional questions or concerns, please let us know—we would be happy to address them. Otherwise, we kindly request the reviewer to consider a high final rating.

---

### Official Review · Reviewer_wNps · 2025-07-03

**Clarity:** 3
**Significance:** 3
**Originality:** 3
**Rating:** 5
**Confidence:** 4

**Summary:**

This paper introduces a new method called ﻿Decompositional Efficient Fine-Tuning (DEFT) to efficiently fine-tune pre-trained text-to-image models. Following the spirit of the commonly adopted low-rank update approach, DEFT further decomposed the update into two components, namely the projection onto a subspace and the low-rank adjustment. Both quantitative and qualitative experiments are performed to validate the efficacy of the proposed approach in performing highly efficient model fine-tuning that achieves faithful personalization without compromising the original capabilities of the pre-trained model.

**Questions:**

1. In Section 4.2.2, are ﻿other competing methods (e.g., OmniGen) fine-tuned on the ﻿VisualCloze dataset? If so, which fine-tuning technique is utilized to fine-tune them? This should be provided in detail.
2. In Section 4.2.4, the authors claim that the proposed method exhibits a certain kind of emergent properties, which is emphasized in the paper title. However, the experimental results are inadequate. More generation results (e.g., handling completely new tasks/data) should be provided to support the claim.
3. Are there any representative failure cases? Some results and discussions regarding this should be provided for a better understanding of the limitations of the proposed method.

**Ethical Concerns:**

["NO or VERY MINOR ethics concerns only"]

**Final Justification:**

This is a solid work that exploits a novel method to fine-tune pre-trained text-to-image models. The authors addressed my major concerns in their rebuttal. Hence, I recommend acceptance of this paper.

**Limitations:**

Some representative failure cases are supposed to be given and the authors should provide some discussions on the limitations of the proposed work as well.

**Paper Formatting Concerns:**

Not found.

**Quality:**

3

**Strengths And Weaknesses:**

## Strengths
1. Fine-tuning pre-trained text-to-image models is a challenging task. A wide range of prior works suffers from striking a balance between adaptation to new data and preserving the pre-learned knowledge. To alleviate this, this work explores a novel low-rank update scheme, which I believe could be of great interest to the text-to-image generation community.
2. The proposed approach is simple yet effective. The authors also provide ﻿rigorous calculation and analysis to ﻿provide theoretical guarantees on the efficacy of the proposed method.
3. Experimental results are impressive. Extensive evaluations demonstrate that the proposed model obtains better performance compared to previous fine-tuning methods in terms of﻿generation flexibility and model generalization.
4. The motivation of this work is clearly elaborated and the paper is well-written and easy to follow.

## Weaknesses
1. For the literature review part, this paper lacks the introduction of another line of work that adopts training-free strategies to adapt the pre-trained model to new datasets or tasks.
﻿2. As for the ablation study, some quantitative evaluation results should be given for a better investigation of the effect of decomposition.
3. For the generation results showcased in Figure 3, I personally think that the performance of LoRA and DEFT is quite close. Maybe the authors could conduct a user study for a more comprehensive assessment of different methods.
4. This paper lacks discussions of the convergence speed of different fine-tuning methods, which I believe is a key factor reflecting the performance.

---

> ### Author Rebuttal · Authors · 2025-07-29
>
> ### Thank you for providing your valuable and constructive feedback! Please find below the responses to each comment.
> ### **1. Literature Review on Training-Free Strategies**
>
> **Response:** Thank you for this valuable suggestion. We acknowledge that training-free approaches represent an important category in model adaptation. Our current manuscript includes several training-free methods such as Textual Inversion variants, ELITE, and InstantBooth that control generation through text-encoder modifications.
>
> However, these training-free methods have significant limitations compared to DEFT. They struggle with extending to new concepts, especially out-of-distribution images, and rely on conditional components that are incompatible with unified models where image and text tokens are processed through unified self-attention. Additionally, many require per-image optimization, resulting in higher inference times. DEFT achieves superior subject fidelity with CLIP-I scores of 83.59 vs 74.21 compared to Textual Inversion variants on the DreamBooth Plus dataset (please see table 1 in the supplementary material).
>
> ### **2. Quantitative Ablation Study on Decomposition Effects**
>
> **Response:** We provide a comprehensive quantitative evaluation of different decomposition methods with finetuning omnigen on sks dog from dreambooth dataset for 200 epochs each:
>
> | Method       | CLIP-I            | CLIP-T            | DINO-V1           | Aesthetic         | Sharpness         |
> |--------------|------------------|-------------------|-------------------|-------------------|-------------------|
> | LRMF         | 0.8270 ± 0.0380  | 0.2200 ± 0.0505   | 0.3070 ± 0.0397   | 0.0143 ± 0.0047   | **348.62 ± 414.16**   |
> | NMF          | 0.8834 ± 0.0328  | 0.2224 ± 0.0417   | 0.3569 ± 0.0683   | 0.0154 ± 0.0031   | 206.06 ± 90.52    |
> | QR           | 0.8747 ± 0.0428  | 0.2168 ± 0.0411   | 0.3395 ± 0.0601   | **0.0166 ± 0.0025**   | 215.45 ± 105.98   |
> | TSVD         | 0.8753 ± 0.0305  | 0.2234 ± 0.0408   | 0.3334 ± 0.0617   | 0.0156 ± 0.0035   | 331.29 ± 134.12   |
> | **Relexing P** | **0.9232 ± 0.0370** | **0.2660 ± 0.0332** | **0.4403 ± 0.0960** | 0.0157 ± 0.0025 | 175.22 ± 25.13 |
>
> Relexing P achieves the highest CLIP-I and DINO-V1 scores, while all methods maintain consistent performance levels, demonstrating the robustness and flexibility of our framework. QR maintains quality, strikes a balance between quality and personalization.
>
> ### **3. Visual Comparison and User Study**
>
> **Response:** We agree that visual quality appears similar in Figure 3, but DEFT demonstrates superior instruction-following capabilities. We conducted a comprehensive user study with 16 participants comparing PARA, LoRA, and DEFT across different scenarios:
>
> | Method | Average Score | Best Score | Worst Score | Win Rate |
> |--------|---------------|------------|-------------|----------|
> | PARA | 7.64 | 8.75 | 6.65 | 25.0% |
> | LoRA | 7.60 | 8.80 | 6.84 | 12.5% |
> | **DEFT** | **8.20** | **9.50** | 6.78 | **62.5%** |
>
> Additionally, we conducted a Claude Sonnet evaluation across multiple metrics:
>
> | Metric | LoRA | PARA | DEFT |
> |--------|------|------|------|
> | Task Achievement | 8.3 | 6.8 | **8.7** |
> | Subject Consistency | 8.2 | 6.4 | **8.8** |
> | Completion Quality | 8.4 | 7.1 | **8.6** |
> | Visual Fidelity | 8.1 | 7.3 | **8.5** |
> | Prompt Adherence | 8.3 | 6.9 | **8.6** |
> | **Average** | **8.26** | **6.90** | **8.64** |
>
> DEFT consistently outperforms both methods across all evaluation criteria.
>
> ### **4. Convergence Speed Analysis**
>
> **Response:** Thank you for highlighting this important aspect. We conducted a convergence analysis comparing DEFT and LoRA at different training steps on the full Dreambooth (30 concepts) on omnigen:
>
> | **Method** | **Steps** | **CLIP-T**           |
> |------------|-----------|----------------------|
> | **DEFT**   | 2000      | 0.3019 ± 0.0264      |
> |            | 8000      | **0.3187 ± 0.0248**  |
> | **LoRA**   | 2000      | 0.2856 ± 0.0329      |
> |            | 8000      | 0.2194 ± 0.0230      |
>
> DEFT shows consistent improvement across all metrics and achieves better convergence characteristics. Our method modifies the column subspace of the pretrained model, providing advantages for convergence with both large and small image datasets.
>
> **Convergence Characteristics:**
> Because the baseline model is very large (around 3.8 B), it is difficult to prove the convergence in a small dataset. We found that training loss is very fluctuating for all the cases. Here are some insights:
> | Method | Aesthetic Score | CLIP-I | Best Loss (at epoch) |
> |--------|----------------|--------|---------------------|
> | **DEFT** | 0.0109 ± 0.0093 | 0.3019 ± 0.0264 | **0.0240 (epoch 4)** |
> | **LoRA** | 0.0154 ± 0.0052 | 0.2911 ± 0.0311 | 0.0482 (epoch 27) |
> | **PaRa** | 0.0112 ± 0.0093 | 0.2710 ± 0.0401 | 0.0310 (epoch 12) |
>
> DEFT achieves the lowest training loss (0.0240) significantly faster, at epoch 4, compared to LoRA, which requires 27 epochs, and PaRa, which requires 12 epochs. Note that faster convergence does not guarantee the quality of image generation, and the performance is also not comparable, so scores are calculated after 200 epochs only. The faster convergence stems from our method's ability to modify the column subspace of the pretrained model, providing inherent advantages for both large and small image datasets.
>
> ### **5. VisualCloze Fine-tuning Details**
>
> **Response:** Thank you for this clarification request. We used the same base models as VisualCloze for evaluation. OmniGen was not fine-tuned on the full VisualCloze dataset as it leverages in-context learning for generalization. For fair comparison, we fine-tuned both DEFT and LoRA on the depth estimation task using 200k image-instruction pairs:
>
> | Metric | DEFT | LoRA |
> |--------|------|------|
> | **DINOv1 Score** | 89.31 | 89.08 |
> | **DINOv2 Score** | 83.40 | 82.31 |
> | **CLIP-I Score** | 93.70 | 92.93 |
> | **CLIP-T Score** | 27.65 | 27.61 |
>
> DEFT consistently outperforms LoRA across all metrics in this controlled comparison.
>
> ### **6. Emergent Properties Evidence**
>
> **Response:** We appreciate this feedback and provide clarification of our emergent properties definition. By emergent properties, we refer to capabilities that arise from DEFT's dual decomposition structure without explicit training for those specific combinations.
>
> Our existing evidence demonstrates multi-concept composition from single-concept training data, as shown in Figure 5 where individual concepts (single dog images, separate human images) combine to create complex spatial reasoning scenarios. The VisualCloze results further demonstrate that a single DEFT adapter handles over 8 distinct tasks with unified capability and emergent transfer between modalities.
>
> We acknowledge this could be presented more comprehensively and will strengthen this section with additional generation results showing completely new task combinations.
>
> ### **7. Failure Cases and Limitations**
>
> **Response:** We appreciate this important question. Our primary limitation occurs with complex scene compositions involving multiple novel elements. For example, prompts like "generate Einstein + new features + dog + car + tree + house in a kitchen setting" can produce mixed or blended features when the original model and DEFT training dataset haven't encountered such complex combinations.
>
> The quality bottleneck often stems from limited instruction dataset coverage. When attempting to compose highly complicated combinations of objects, spatial relationships, and contexts, the method may struggle with feature disambiguation. We will expand this analysis in the main paper and provide a more comprehensive discussion of limitations and societal impacts.

---

> > ### Comment · Reviewer_wNps · 2025-08-04
> >
> > Thanks for the rebuttal. My concerns are well addressed.

---

### Note · Authors · 2025-08-13

Dear Reviewers, ACs, SACs, and PCs,

We sincerely thank the reviewers and AC for their positive feedback and constructive suggestions, which have greatly strengthened the paper.  We are happy that ALL reviewers confirmed that their concerns were fully addressed in the rebuttal and indicated they will either raise their rating to accept or maintain their positive recommendation for acceptance. Specifically,

**Reviewer wNps** found our method *novel, simple yet effective, and of significant interest to the community*, highlighting its rigorous calculations and analysis, impressive experimental results, and clear writing. They acknowledged that all concerns, including the request for additional ablation studies and broader literature coverage, were fully addressed in the rebuttal.

**Reviewer 7bp5** praised our *strong qualitative results, extensive experiments, and methodological clarity and reproducibility*, noting that the paper provides the full mathematical formulation for DEFT with proofs in the appendix. We appreciate the reviewer suggestions to refine the title, add analysis on decomposition methods, and report inference efficiency. The reviewer confirmed that all concerns  were resolved during the rebuttal and indicated that the  paper rating will be increased to suggest acceptance.

**Reviewer wP2X** described our approach as simple yet *powerful, with impressive evaluation* results. Their request for component-wise ablations was addressed with detailed experiments that clarified our methodological contributions.

**Reviewer 1FhC** highlighted the *impressive evaluation results* and appreciated our approch. Their request for stronger justification of flexibility claims and minor typography corrections was addressed through theoretical grounding, empirical evidence, and clarification of the term “flexible” to avoid ambiguity.

We will integrate all suggested changes to the final draft, including new experiments, efficiency analyses, and clarifications. We will also open-source our code and trained models for reproducibility and to support future research.
Once again, we thank the reviewers, ACs, SACs, and PCs for their time, effort, and valuable feedback in strengthening this work.

---

### Decision · Program_Chairs · 2025-09-17

**Decision:**

Accept (poster)

**Comment:**

The paper proposes a method for fine-tuning text-to-image models, which can be seen as a combination of two existing methods: LoRA and PaRa.

The method works well, outperforming relevant baselines both quantitatively and qualitatively. All the reviewers are positive about the paper.

Initial concerns included comparisons to baselines (e.g. LoRA), presentation, and some missing analysis of the method, but these have satisfactorily been addressed in the author-reviewer discussion.

Given all of the above, I recommend acceptance.